American Society for Microbiology | Microbiology Spectrum

# Long-Term and Low-Level Envelope C2V3 Stimulation by Highly Diverse Virus Isolates Leads to Frequent Development of Broad and Elite Antibody Neutralization in HIV-1-Infected Individuals

Francisco Martin,[a] José Maria Marcelino,[a,b] Claudia Palladino,[a] Inês Bártolo,[a] Susana Tracana,[a] Inês Moranguinho,[a] Paloma Gonçalves,[a] Rita Mateus,[a] Rita Calado,[a] Pedro Borrego,[a] Thomas Leitner,[c] Sofia Clemente,[d] Nuno Taveira[a,b]

[a]Research Institute for Medicine, Faculty of Pharmacy, Universidade de Lisboa, Lisbon, Portugal
[b]Centro de Investigação Interdisciplinar Egas Moniz, Instituto Universitário Egas Moniz, Caparica, Portugal
[c]Theoretical Biology and Biophysics Group, Los Alamos National Laboratory, Los Alamos, New Mexico, USA
[d]Hospital da Divina Providência, Luanda, Angola

**ABSTRACT** A minority of HIV-1-infected patients produce broadly neutralizing antibodies (bNAbs). Identification of viral and host correlates of bNAb production may help develop vaccines. We aimed to characterize the neutralizing response and viral and host-associated factors in Angola, which has one of the oldest, most dynamic, and most diverse HIV-1 epidemics in the world. Three hundred twenty-two HIV-1-infected adults from Angola were included in this retrospective study. Phylogenetic analysis of C2V3C3 *env* gene sequences was used for virus subtyping. Env-binding antibody reactivity was tested against polypeptides comprising the C2, V3, and C3 regions. Neutralizing-antibody responses were determined against a reference panel of tier 2 Env pseudoviruses in TZM-bl cells; neutralizing epitope specificities were predicted using ClustVis. All subtypes were found, along with untypeable strains and recombinant forms. Notably, 56% of the patients developed cross neutralizing, broadly neutralizing, or elite neutralizing responses. Broad and elite neutralization was associated with longer infection time, subtype C, lower CD4$^+$ T cell counts, higher age, and higher titer of C2V3C3-specific antibodies relative to failure to develop bNAbs. Neutralizing antibodies targeted the V3-glycan supersite in most patients. V3 and C3 regions were significantly less variable in elite neutralizers than in weak neutralizers and nonneutralizers, suggesting an active role of V3C3-directed bNAbs in controlling HIV-1 replication and diversification. In conclusion, prolonged and low-level envelope V3C3 stimulation by highly diverse and ancestral HIV-1 isolates promotes the frequent elicitation of bNAbs. These results provide important clues for the development of an effective HIV-1 vaccine.

**IMPORTANCE** Studies on neutralization by antibodies and their determinants in HIV-1-infected individuals have mostly been conducted in relatively recent epidemics caused by subtype B and C viruses. Results have suggested that elicitation of broadly neutralizing antibodies (bNAbs) is uncommon. The mechanisms underlying the elicitation of bNAbs are still largely unknown. We performed the first characterization of the plasma neutralizing response in a cohort of HIV-1-infected patients from Angola. Angola is characterized by an old and dynamic epidemic caused by highly diverse HIV-1 variants. Remarkably, more than half of the patients produced bNAbs, mostly targeting the V3-glycan supersite in HIV-1. This was associated with higher age, longer infection time, lower CD4$^+$ T cell counts, subtype C infection, or higher titer of C2V3C3-specific antibodies relative to patients that did not develop bNAbs. These results may help develop the next generation of vaccine candidates for HIV-1.

**KEYWORDS** Angola, Env diversity, HIV-1 infection, broadly neutralizing antibodies, bNAbs, Env-specific antibodies, neutralizing epitopes

Address correspondence to Nuno Taveira, ntaveira@ff.ul.pt.

The authors declare no conflict of interest.

The HIV-1 envelope glycoprotein is highly immunogenic, and 5% to 50% of HIV-1-infected adults develop broadly neutralizing antibodies (bNAbs) after several years of infection (1–7). These bNAbs have little impact on the control of the infection due to the capacity of autologous isolates to continuously diversify and escape these antibodies (4, 8, 9). However, some recombinant human bNAbs suppress viral replication in HIV-1-infected individuals (10–15) and prevent human infection by some HIV-1 strains (16), and passive immunization in animal models can protect against infection and/or disease progression (17). Therefore, bNAbs are promising tools to restrict HIV-1 transmission and control disease progression if they can be induced by vaccination. Unfortunately, so far, antibodies elicited by candidate immunogens and vaccines have shown a limited ability to neutralize heterologous primary HIV-1 strains (8, 18–24).

Most bNAbs target five highly conserved epitopes in the HIV-1 envelope: the CD4 binding site (CD4bs); the N-linked glycan at position 160 in the V2 apex; the V3 glycan supersite; the gp41 membrane external proximal region (MPER); and the gp41/gp120 interface, which includes the fusion peptide (17, 25–28). Guiding the immune system to produce such bNAbs remains a major challenge due to the extremely complex antibody maturation pathways and high levels of somatic hypermutation required by HIV-1-specific antibodies to acquire neutralization breadth (28–30). Exceptions are some V3-glycan supersite bNAbs that do not require extensive antibody affinity maturation (28, 31, 32), allowing their development in early stages of infection (32, 33) and explaining their high prevalence in recently infected individuals (1). Such findings, together with the proved therapeutic value of V3-glycan supersite bNAbs (12), highlight this epitope as a key target for HIV vaccine design.

Understanding the determinants of bNAb production in some HIV-1-infected individuals is of crucial importance for the development of improved immunogens and immunization strategies. In Switzerland, data analysis by the Swiss HIV Cohort indicated that ethnicity was associated with bNAb induction, with black participants being more prone to develop bNAb responses (3). Regarding the impact of HIV-1 subtype in plasma neutralization, some studies found limited to no impact of HIV-1 subtype in plasma neutralization, suggesting that HIV-1 group M subtypes and neutralization response evolved independently (4, 34, 35). More recently, in a large longitudinal sub-Saharan HIV primary infection cohort, cross-clade plasma neutralization was strongly correlated with subtype C infection (1). Additionally, Rusert et al. (3) found a strong association between plasma neutralization specificity and HIV-1 subtype, with subtype B viruses being more vulnerable to CD4 binding site-specific antibodies and non-B viruses being more vulnerable to V2-glycan-specific neutralizing antibodies. We recently showed that the frequency and level of antibody response to selected epitopes in the envelope gp41 differ between HIV-1-infected patients from Germany, France, and Portugal, which have different subtype distributions (36).

Considering that vaccine effectiveness will depend on the extent to which induced antibodies will neutralize the diverse HIV-1 variants circulating globally, it is important to characterize HIV-1 antibody responses in different epidemics and geographies. The neutralizing-antibody response of HIV-1-infected patients from Angola has never been evaluated. HIV-1 was introduced into Angola from Kinshasa, the capital city of the Democratic Republic of Congo (DRC), in 1910 to 1940 (37). As in DRC, all subtypes, multiple circulatory recombinant forms (CRFs) and unique recombinant forms (URFs) are present in Angola along with highly divergent and ancestral forms of the different subtypes (37–42). The genetic complexity of the HIV-1 quasispecies present in infected individuals is directly related to the development of neutralization breadth regardless of infection duration (43, 44). This should be particularly evident in old and dynamic epidemics such as the one in Angola. Here, we carried out the first detailed characterization of the neutralizing-antibody response against HIV-1 in Angola and identified viral and host factors associated with the neutralizing response.

## RESULTS

**Characterization of the study population and infecting HIV-1 isolates.** Overall, 375 plasma samples from 322 adult HIV-1-infected patients from three sampling years, 2001 ($n = 106$), 2009 ($n = 210$), and 2014 ($n = 59$), were included in this analysis. Epidemiological, clinical, demographic, and virological characterization of the patients is provided in Table S1 in the supplemental material. The median age of the patients was 34 years, and most (242 [75.2%]) were women. The main route of transmission was heterosexual contact (304 [94.4%]). There were no significant differences related to age and sex between sampling years. The median plasma viral load (VL) at the time of sampling was significantly higher in 2001 than 2009 (4.2-fold higher) and 2014 (33.5-fold higher). The median number of CD4$^+$ T cells in 2014 was 1.8-fold higher than in 2009 ($P = 0.0015$). The lower VL and higher CD4$^+$ T cell number in 2014 are consistent with most patients being on combination antiretroviral therapy (cART), which was not the case in 2001 and 2009.

Sequencing and phylogenetic analysis of the C2V3C3 Env region (comprising the envelope regions C2, V3, and C3) were completed successfully for 206 patients from 2001 (96/106 [90.6%]) and 2009 (110/210 [52.4%]). The following subtypes were identified: A1 (2001, $n = 33$ [34.4%]; 2009, $n = 32$ [29.1%]), A2 (2001, $n = 6$ [6.3%]; 2009, $n = 3$ [2.7%]), B (2001, $n = 2$ [2.1%]; 2009, $n = 2$ [1.8%]), C (2001, $n = 12$ [12.5%]; 2009, $n = 30$ [27.3%]), D (2001, $n = 2$ [2.1%]; 2009, $n = 8$ [7.3%]), F1 (2001, $n = 5$ [5.2%]; 2009, $n = 6$ [5.5%]), G (2001, $n = 8$ [8.3%]; 2009, $n = 11$ [10%]), H (2001, $n = 19$ [19.8%]; 2009, $n = 15$ [13.6%]), and J (2001, $n = 3$ [3.1%]; 2009, $n = 0$ [0.0%]). Untypeable (U) strains were found at rates of 4.2% ($n = 6$) in 2001 and 2.7% ($n = 3$) in 2009 (Fig. S1). Subtype A prevailed in 2001 and 2009, but subtype C increased significantly (2.2-fold; $P = 0.0095$) in 2009. Of the 176 isolates for which there were protease (PR) and C2V3C3 sequences available, 74 (42.0%) were nonrecombinant and 102 (58.0%) were recombinant. Recombinant strains prevailed over pure subtypes in 2001 and 2009 (Table S2). Unfortunately, we could not sequence the C2V3C3 region from most of the 2014 samples due to their low or undetectable viral load (Table S1). Moreover, the lack of plasma prevented further analysis of samples collected in 2001.

**Characterization of the antibody response.** A total of 236 plasma samples, 178 from 2009 and 58 from 2014, were screened for neutralization breadth and potency against the 12 Env-pseudotyped viruses in the indicator panel (Fig. 1 and Fig. S2). In 2009, 80.9% (144/178) of Angolan patients effectively neutralized at least one virus in the indicator panel; this increased to 93.1% (54/58) in 2014 (Fig. 1). The overall neutralization potency, calculated as percent neutralization of all tested variants at a 1:40 plasma dilution (27.43% with a 95% confidence interval [CI] of 25.94 to 28.92 in 2009 versus 60.52% [95% CI, 57.37 to 63.66] in 2014; $P < 0.0001$), and the mean neutralization breadth (4.39 [95% CI, 3.83 to 4.95] in 2009 versus 8.40 [95% CI, 7.32 to 9.48] in 2014; $P < 0.0001$) were higher in 2014 than 2009.

The mean neutralization score (NS) (see Materials and Methods) was 11.71 (95% CI, 10.22 to 13.19), ranging from 0 to 36, and the median was 7 (interquartile range [IQR], 2.0 to 21.0). Remarkably, approximately 30% (68/236) of the plasma samples tested from the Angolan patients had antibody responses with the capacity to potently neutralize at least half the viruses in the panel (Fig. 2A and B). Overall, considering both sampling years, 18.6% (44/236) of study participants were elite neutralizers (NS $\geq 25$), 10.2% (24/236) were broad neutralizers ($18 \leq$ NS $< 25$), 27.1% (64/236) were cross neutralizers ($6 \leq$ NS $< 17$), and 44.1% (104/236) were weak neutralizers or did not neutralize any virus in the panel (Fig. 2B).

**Correlates of the neutralizing response.** The neutralizing-antibody response has been previously associated with viral load, CD4$^+$ T cell count, viral diversity, and infection time (1, 3). We first analyzed the impact of sample collection time on the neutralizing-antibody responses of the HIV-infected Angolan patients. Strikingly, median NS was 6.2-fold higher in 2014 than 2009 (31.00 [IQR, 10.50 to 33.00] versus 5.00 [IQR, 1.00 to 13.25]; $P < 0.0001$) (Fig. 2A). Consistent with this, the frequency of elite neutralizers was 9.5-fold higher in 2014 than in 2009 (57% [33/58] versus 6% [11/178]; $P < 0.0001$),

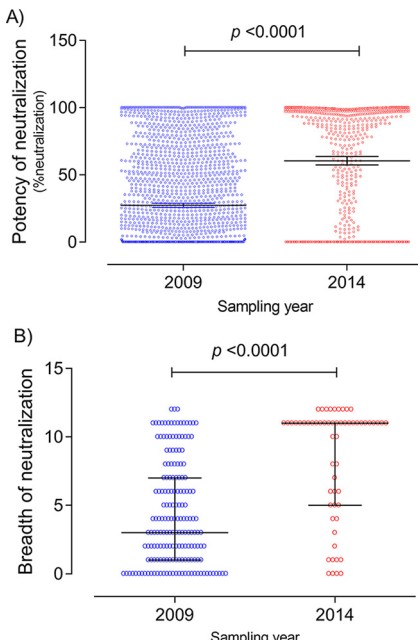

**FIG 1** Neutralization potency and breadth per sampling year. (A) Potency of neutralization (percent neutralization at a 1:40 plasma dilution) of samples collected in 2009 and 2014 as assessed against the 12 Env-pseudotyped viruses in the indicator panel. Means and 95% confidence intervals are shown. (B) Neutralization breadth (number of Env-pseudotyped viruses neutralized at >20%) in samples collected in 2009 and 2014. Medians and interquartile ranges are shown. $P$ values were obtained using the Mann-Whitney U test.

and weak neutralizers and nonneutralizers were 2.7-fold more frequent in 2009 than in 2014 (52% [93/178] versus 19.0% [11/58]; $P < 0.0001$) (Fig. 2B). Broad neutralizers were 2.4-fold more frequent in 2009 than in 2014 (12% [21/178] versus 5% [3/58]; $P = 0.2107$), and a similar trend was observed for cross neutralizers (30% [53/179] versus 19.0% [11/58]; $P = 0.1270$). We also analyzed matching plasma pairs from 2009 and 2014 to determine the evolution of neutralizing-antibody response as a function of infection time. In line with the previous results, neutralizing score increased in 2014 relative to 2009 in 31 of the 38 matched plasma pairs analyzed (81.6%). (Fig. 2C). The NS was unrelated to the sex of the patients (Fig. 2D). The significant increase in neutralizing score in 2014 relative to 2009 in matched samples suggests that higher duration of infection contributes to increased potency and breadth of the neutralizing-antibody response in Angolan patients infected with HIV-1.

The inhibitory dilution 50% neutralization titers ($ID_{50}$) against the 12-Env-pseudotyped-virus indicator panel were determined in a subset of plasma samples from 2009 ($n = 28$) and 2014 ($n = 10$) showing broad and elite neutralizing activity (Fig. 3A). When unmatched samples were compared, neutralization titers were significantly higher in 2014 than in 2009 (median $\log_{10} ID_{50}$ in 2009, 1.903 [IQR, 1.602 to 2.505; 336 plasma-virus pairs]; median $\log_{10} ID_{50}$ in 2014, 2.204 [IQR, 1.903 to 2.806; 120 plasma-virus pairs]; $P = 0.0013$) (Fig. 3B).

To analyze the impact of HIV-1 subtype on neutralization by Angolan samples we compared the NS in patients infected with subtypes C ($n = 27$) and A1 ($n = 26$), the two prevailing subtypes in Angola, and in patients infected with the other subtypes and recombinant forms ($n = 56$). Only samples collected in 2009 were included in this analysis due to the limited number of samples genotyped in 2014. NS varied significantly with infecting virus subtype ($P = 0.014$), with subtype C leading to significantly higher NS than subtype A1 (median NS = 17.00 [IQR, 6.00 to 25.00] versus 6.00 [IQR, 3.50 to 15.00]; $P = 0.0103$) or other subtypes (median NS = 17.00 [IQR, 6.00 to 25.00] versus 8.00 [IQR, 1.00 to 17.75]; $P = 0.0087$) (Fig. 4A). These results indicate that in 2009, virus

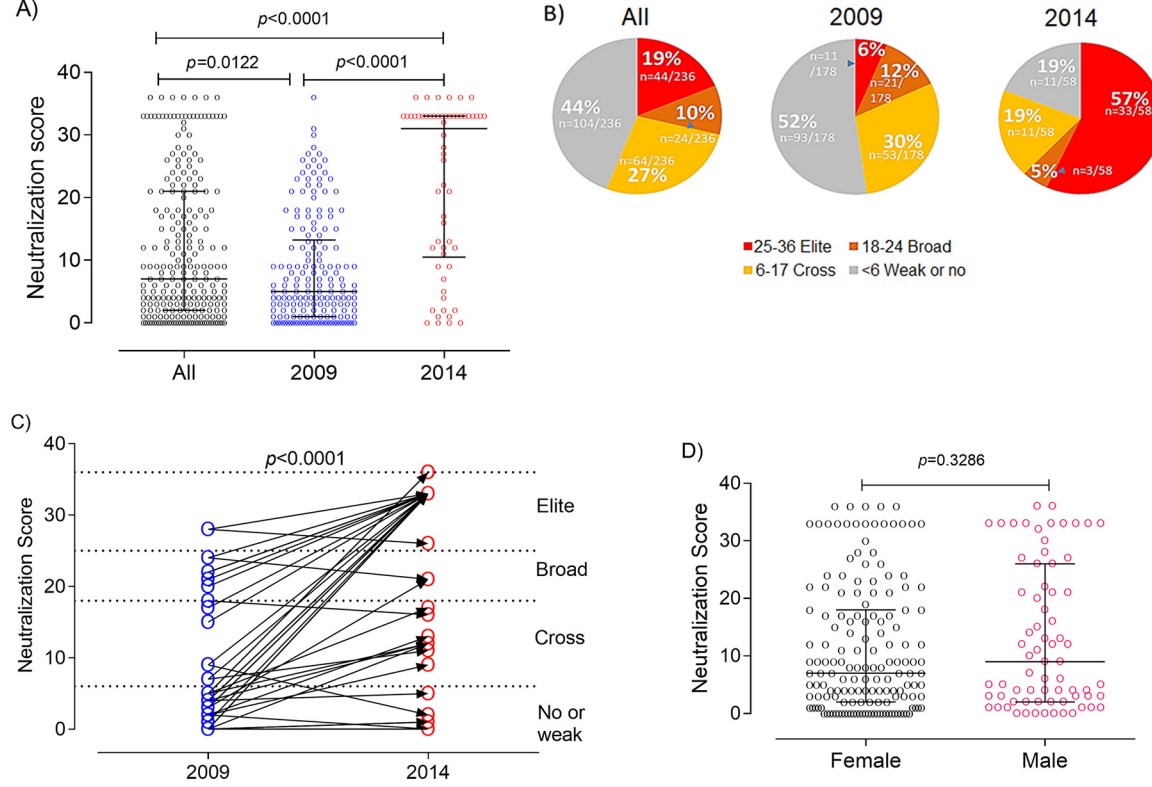

**FIG 2** NSs of samples from HIV-1-infected patients from Angola as a function of year of sampling and sex. (A) NSs in 2009 is represented in blue and in 2014 in red; NSs for all patients is in green. (B) Angolan patients from 2009 and 2014 were categorized into 4 groups according to the NS as follows: nonneutralizers or weak neutralizers, <6 (gray); cross neutralizers, 6 to 17 (yellow); broad neutralizers, 18 to 24 (orange); elite neutralizers, 25 to 36 (red). (C) NS in matched samples collected in 2009 and 2014, showing the NS categories. (D) NS in males and females. Medians and interquartile ranges are shown. *P* values were obtained using the Mann-Whitney U test.

subtype was a major determinant of the neutralizing-antibody response in our HIV-1-infected Angolan patients.

The indicator virus panel used in the neutralization experiments contains three subtype C strains (25710, CE1176, and CE0217) that could be more closely related to the subtype C isolates from the Angolan patients and explain the higher NS observed in patients infected with subtype C viruses. To examine this issue, we compared the susceptibility of the reference panel isolates to neutralization and found a significant variation related to virus subtype (Fig. 4B). The easiest viruses to neutralize were isolate 25710 (subtype C; India) and 398F1 (subtype A; Tanzania). On the other hand, viruses most resistant to neutralization were 2278 (subtype B; Spain) and CNE8 (CRF01_AE; China).

To investigate the impact of the evolutionary distance between the HIV-1 Angolan isolates and the indicator virus panel on the neutralizing-antibody responses, we aligned the C2V3C3 amino acid sequences from the Angolan isolates (in 2009) with those from the indicator virus panel. As expected, subtype C viruses from the patients were more closely related to subtype C viruses of the indicator panel than to other subtypes (Fig. S3A). There was a significant negative correlation of amino acid distance of the indicator panel to NS (Spearman $r = -0.2319$; $P = 0.019$) (Fig. S3B). Hence, the closer the isolate from the indicator panel was to the patient's C2V3C3 amino acid sequence, the more easily it was neutralized. On average, clade C reference strain 25710 from the indicator panel was the closest indicator virus panel member to the Angolan isolates, and, not surprisingly, it was the easiest virus to neutralize. At the other end of the spectrum, clade B reference strain 2278 was the furthest from the C2V3C3 Angolan sequences and was the most difficult virus to neutralize, along with the CRF01_AE virus (CNE8). Nevertheless, many patients infected with all subtypes developed potent bNAb responses despite the

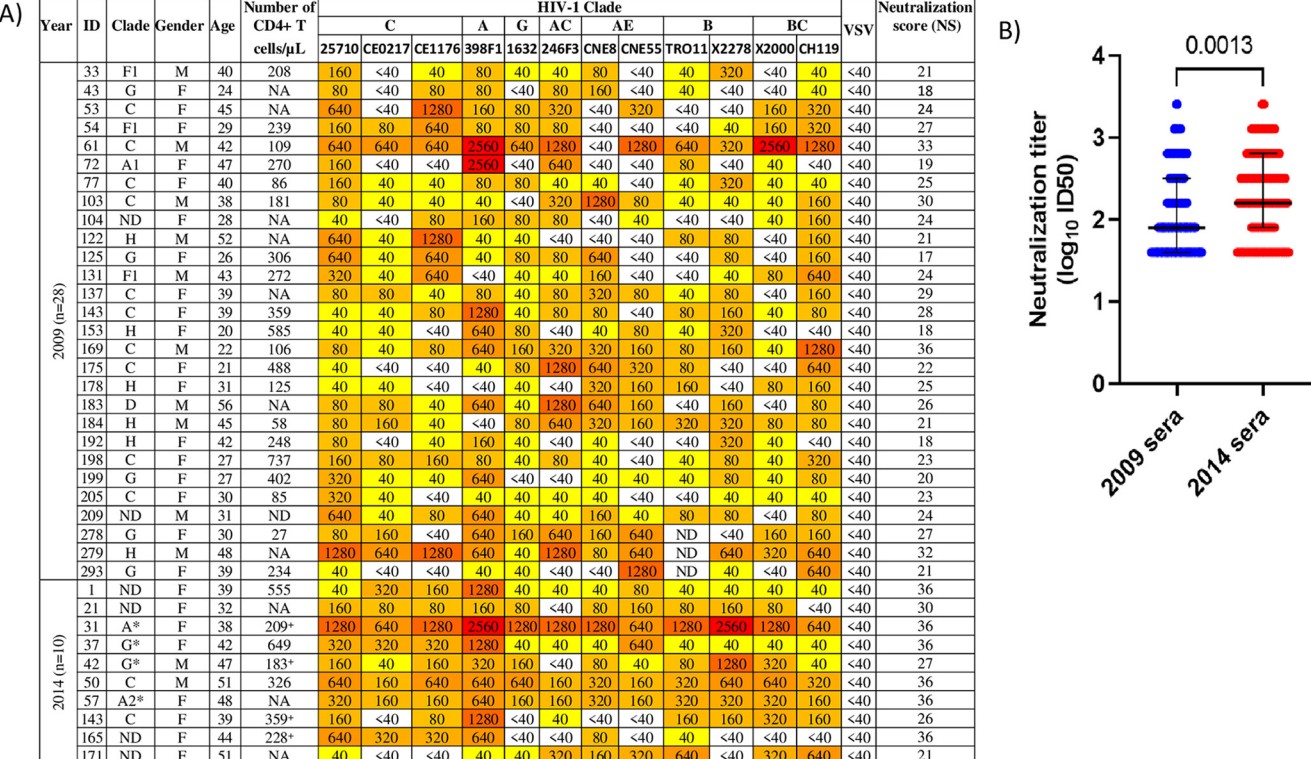

| Year | ID | Clade | Gender | Age | Number of CD4+ T cells/μL | C 25710 | C CE0217 | C CE1176 | A 398F1 | G 1632 | AC 246F3 | AE CNE8 | AE CNE55 | B TRO11 | B X2278 | BC X2000 | BC CH119 | VSV | Neutralization score (NS) |
|---|---|---|---|---|---|---|---|---|---|---|---|---|---|---|---|---|---|---|---|
| 2009 (n=28) | 33 | F1 | M | 40 | 208 | 160 | <40 | 40 | 80 | 40 | 40 | 80 | <40 | 40 | 320 | <40 | 40 | <40 | 21 |
| | 43 | G | F | 24 | NA | 80 | <40 | 80 | 80 | <40 | 80 | 160 | <40 | 40 | <40 | <40 | 40 | <40 | 18 |
| | 53 | C | F | 45 | NA | 640 | <40 | 1280 | 160 | 80 | 320 | <40 | 320 | <40 | <40 | 160 | 320 | <40 | 24 |
| | 54 | F1 | F | 29 | 239 | 160 | 80 | 640 | 80 | 80 | 80 | <40 | <40 | <40 | 40 | 160 | 320 | <40 | 27 |
| | 61 | C | M | 42 | 109 | 640 | 640 | 640 | 2560 | 640 | 1280 | 640 | 1280 | 640 | 320 | 2560 | 1280 | <40 | 33 |
| | 72 | A1 | F | 47 | 270 | 160 | <40 | <40 | 2560 | <40 | 640 | <40 | <40 | 80 | <40 | 40 | <40 | <40 | 19 |
| | 77 | C | F | 40 | 86 | 160 | 40 | 40 | 80 | 80 | 40 | 40 | <40 | 40 | 320 | 40 | 40 | <40 | 25 |
| | 103 | C | M | 38 | 181 | 80 | <40 | 40 | 40 | <40 | 320 | 1280 | 80 | 40 | 40 | 40 | 160 | <40 | 30 |
| | 104 | ND | F | 28 | NA | 40 | <40 | 80 | 160 | 80 | 80 | <40 | 40 | <40 | <40 | 40 | 160 | <40 | 24 |
| | 122 | H | M | 52 | NA | 640 | 40 | 1280 | 40 | 40 | <40 | <40 | <40 | 80 | 80 | <40 | 160 | <40 | 21 |
| | 125 | G | F | 26 | 306 | 640 | 40 | 640 | 40 | 80 | 80 | 640 | <40 | <40 | 80 | <40 | 160 | <40 | 17 |
| | 131 | F1 | M | 43 | 272 | 320 | 40 | 640 | <40 | 40 | 40 | 160 | <40 | 40 | 40 | 80 | 640 | <40 | 24 |
| | 137 | C | F | 39 | NA | 80 | 80 | 40 | 80 | 40 | 80 | 320 | 80 | 40 | 80 | <40 | 160 | <40 | 29 |
| | 143 | C | F | 39 | 359 | 40 | 40 | 40 | 1280 | 40 | 80 | 80 | <40 | 80 | 160 | 40 | 80 | <40 | 28 |
| | 153 | H | F | 20 | 585 | 40 | 40 | <40 | 640 | 80 | <40 | 40 | 80 | 40 | 320 | <40 | <40 | <40 | 18 |
| | 169 | C | M | 22 | 106 | 80 | 40 | 80 | 640 | 160 | 320 | 320 | 160 | 80 | 160 | 40 | 1280 | <40 | 36 |
| | 175 | C | F | 21 | 488 | 40 | <40 | <40 | 40 | 80 | 1280 | 640 | 320 | 80 | <40 | <40 | 640 | <40 | 22 |
| | 178 | H | F | 31 | 125 | 40 | 40 | <40 | <40 | 40 | <40 | 320 | 160 | 160 | <40 | 80 | 160 | <40 | 25 |
| | 183 | D | M | 56 | NA | 80 | 80 | 40 | 640 | 40 | 1280 | 640 | 160 | <40 | 160 | <40 | 80 | <40 | 26 |
| | 184 | H | M | 45 | 58 | 80 | 160 | 40 | <40 | 80 | 640 | 320 | 160 | 320 | 320 | 80 | 80 | <40 | 21 |
| | 192 | H | F | 42 | 248 | 80 | <40 | 40 | 160 | 40 | <40 | 40 | <40 | <40 | 320 | 40 | <40 | <40 | 18 |
| | 198 | C | F | 27 | 737 | 160 | 80 | 160 | 80 | 40 | 80 | 40 | <40 | 40 | 40 | 40 | 320 | <40 | 23 |
| | 199 | G | F | 27 | 402 | 320 | 40 | 40 | 640 | <40 | <40 | 40 | 40 | 40 | 80 | 40 | 80 | <40 | 20 |
| | 205 | C | F | 30 | 85 | 320 | 40 | <40 | 40 | 40 | 40 | 40 | <40 | <40 | 40 | 40 | 40 | <40 | 23 |
| | 209 | ND | M | 31 | ND | 640 | 40 | 80 | 640 | 40 | 40 | 160 | 40 | 80 | <40 | 80 | 40 | <40 | 24 |
| | 278 | G | F | 30 | 27 | 80 | 160 | <40 | 640 | 160 | 640 | 160 | 640 | ND | <40 | 160 | 160 | <40 | 27 |
| | 279 | H | M | 48 | NA | 1280 | 640 | 1280 | 640 | 40 | 1280 | 80 | 640 | ND | 640 | 320 | 640 | <40 | 32 |
| | 293 | G | F | 39 | 234 | 40 | <40 | <40 | 40 | 40 | <40 | <40 | 1280 | ND | <40 | 640 | 640 | <40 | 21 |
| 2014 (n=10) | 1 | ND | F | 39 | 555 | 40 | 320 | 160 | 1280 | 40 | 40 | 40 | 80 | 40 | 40 | 40 | 40 | <40 | 36 |
| | 21 | ND | F | 32 | NA | 160 | 80 | 80 | 160 | 80 | <40 | 80 | 160 | 80 | 160 | 80 | <40 | <40 | 30 |
| | 31 | A* | F | 38 | 209+ | 1280 | 640 | 1280 | 2560 | 1280 | 1280 | 1280 | 640 | 1280 | 2560 | 1280 | 640 | <40 | 36 |
| | 37 | G* | F | 42 | 649 | 320 | 320 | 320 | 1280 | 40 | 40 | 40 | 640 | 40 | 40 | 40 | 40 | <40 | 36 |
| | 42 | G* | M | 47 | 183+ | 160 | 40 | 160 | 320 | 160 | <40 | 80 | 40 | 80 | 1280 | 320 | 40 | <40 | 27 |
| | 50 | C | M | 51 | 326 | 640 | 160 | 640 | 640 | 640 | 160 | 320 | 640 | 320 | 640 | 640 | 320 | <40 | 36 |
| | 57 | A2* | F | 48 | NA | 320 | 160 | 160 | 640 | 160 | 160 | 320 | 160 | 320 | 320 | 320 | 160 | <40 | 36 |
| | 143 | C | F | 39 | 359+ | 160 | <40 | 80 | 1280 | <40 | 40 | <40 | <40 | 160 | 160 | 320 | 160 | <40 | 26 |
| | 165 | ND | F | 44 | 228+ | 640 | 320 | 320 | 640 | <40 | <40 | 80 | <40 | 40 | <40 | <40 | <40 | <40 | 36 |
| | 171 | ND | F | 51 | NA | 40 | <40 | <40 | 40 | 40 | 320 | 160 | 320 | 640 | <40 | 320 | 640 | <40 | 21 |

**FIG 3** Antibody neutralization titers in a subset of plasma samples from elite and broad neutralizers from 2009 (n = 28) and 2014 (n = 10). (A) Heat map of the neutralization titers (ID$_{50}$) and neutralization scores against the 12 Env-pseudotyped-virus indicator panel. ID$_{50}$s are color coded, with darker colors indicating higher ID$_{50}$s. Subtype was determined in the *env* gene except for samples noted with asterisks, where subtype was determined in the *pol* gene. +, number of CD4$^+$ T cells determined in 2009; ND, not done due to lack of sample; NA, not available. VSV-pseudotyped viruses were used as a neutralization specificity control. ID$_{50}$ heat map color code: white, <40; bright yellow, 40; yellow, 80 to 320; orange, 640 to 1,280; red, 2,560. (B) Comparison of antibody neutralization titers in 2009 and 2014. Log$_{10}$ ID$_{50}$s obtained with each patient sample against the 12 Env-pseudotyped viruses in the indicator panel are plotted. Lines indicate medians with interquartile ranges. The *P* value was obtained using the Mann-Whitney U test.

great genetic distance from the viruses of the indicator panel, indicating that other factors besides the relatedness to the indicator panel strains contribute to the potency of the neutralizing response.

Drug-naive patients (in 2009) with ≤200 CD4$^+$ T cells/μL at study entry had significantly higher NS values than patients with >200 CD4$^+$ T cells/μL (median NS in patients with ≤200 CD4$^+$ T cells/μL was 7.00 [IQR, 3.50 to 21.00] versus 4.00 [IQR, 1.00 to 12.00] in patients with >200 CD4$^+$ T cells/μL; *P* = 0.0193) (Fig. 5A). Moreover, NS values were inversely associated with CD4$^+$ T cell counts (Spearman *r* = −0.3043; *P* = 0.0005) (Fig. 5B) and directly associated with age (Spearman *r* = 0.1644; *P* = 0.0302) in these patients (Fig. 5C). These results suggest that elicitation of high levels of broadly neutralizing antibodies in these patients is directly related to prolonged antigenic stimulation (45).

**Epitope specificities of the plasma neutralizing antibodies.** In the same subset of 38 plasma samples (28 from 2009 and 10 from 2014) from broad and elite neutralizers, the epitope specificities were mapped using a computational clustering tool based on the epitope specificities of a panel of human bNAbs (46). Six (15.8%) samples did not cluster with any of the bNAbs. Thirty-two (84.2%) samples clustered with one of the bNAbs. Of these samples, most (68.8% [22/32], of which 16 were from 2009 and 6 from 2014) clustered with PGT128 and 2G12, two bNAbs that target the V3 glycan supersite with important contact residues in V3 and V4 (Fig. 6) (47, 48). Five (15.6%) samples clustered with bNAb 4E10, which targets the gp41 membrane-proximal external region (MPER) (49). Four (12.5%) samples clustered with VRC01 and VRC-CH31, which target the CD4 binding site. Finally, one (3.1%) sample clustered with PG16 and PG9, which target the V1V2 glycans. These results indicate that the V3 glycan supersite is the dominant broadly neutralizing epitope in Angolan patients.

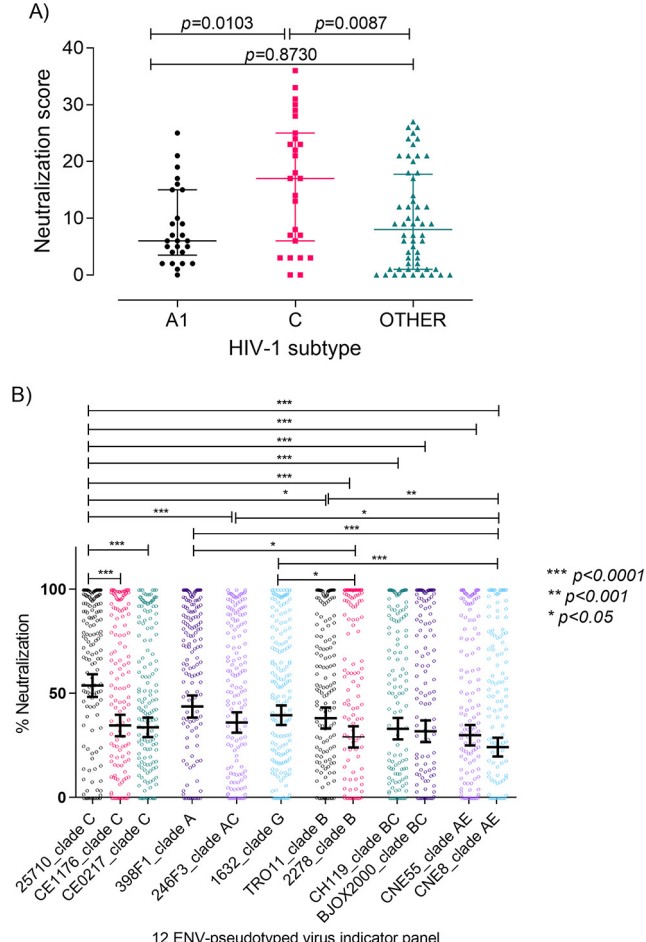

**FIG 4** Impact of HIV-1 subtype on antibody neutralization. (A) Neutralization score in patients infected with the two most common subtypes in Angola (in 2009), C (*n* = 27) and A1 (*n* = 26), and in patients infected with other subtypes and recombinant forms (*n* = 56). The Kruskal-Wallis nonparametric test was used to analyze the difference in median NS for all subtypes (*P* = 0.014). Dunn's multiple-comparison test was used to analyze differences in NS between subtypes. Medians and interquartile ranges are shown. (B) Percent neutralization of each of the 12 Env-pseudotyped viruses in the indicator panel by the plasma samples (at a 1:40 dilution) from the Angolan patients (*n* = 236). Mean percent neutralization and 95% confidence interval bars against a given virus from the indicator panel are shown. Statistically significant differences are represented by the *P* values obtained with Dunn's multiple-comparison test. ***, *P* < 0.0001; **, *P* < 0.001; *, *P* < 0.05.

**Neutralization score is directly related to the titer of C2V3C3-binding antibodies in all subtypes.** The antibody binding reactivity against a panel of recombinant polypeptides comprising the C2, V3, and C3 envelope regions of subtypes B, C, G, H, J, and CRF02_AG was characterized in a subset of samples from 2009 (*n* = 48) and 2014 (*n* = 16) with known antibody neutralization profiles. All but the B polypeptide were derived from Angolan isolates. All but six samples from five patients reacted with all C2V3C3 polypeptides, demonstrating the high antigenicity of this envelope region (Fig. S4). In 2009, patients had significantly higher median antibody binding titers against subtype C than against subtypes G (*P* = 0.0007), H (*P* = 0.0282), J (*P* = 0.0052), and CRF02_AG (*P* = 0.0149). Of note, median antibody binding titers were always higher in 2014 than in 2009 regardless of the C2V3C3 polypeptide subtype, but this was not significant except for CRF02_AG.

The higher antibody reactivity against subtype C antigen could be related with the higher neutralizing responses observed in subtype C-infected patients. We therefore investigated possible associations between neutralization score, C2V3C3 antibody binding titer, and subtype. Remarkably, C2V3C3 antibody binding titer was positively associated with NS values; i.e., patients with higher titers of antibody binding to C2V3C3 polypeptides had

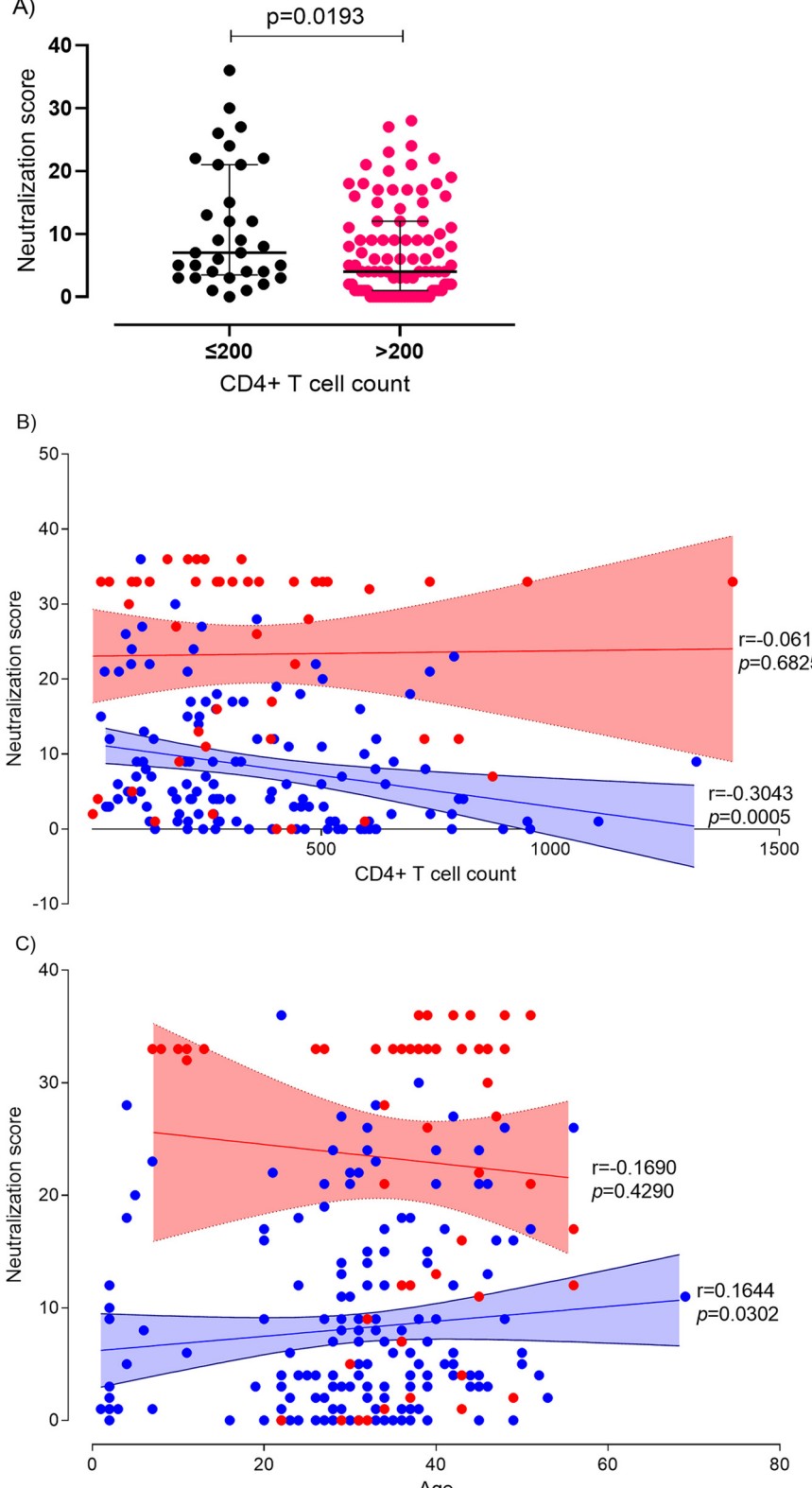

**FIG 5** Correlation between neutralization score, CD4$^+$ T cell counts, and patient's age. (A) Neutralization score differences between 2009 patients with CD4$^+$ T cell counts of ≤200/µL at study entry and patients with counts of >200/µL. Median and interquartile range are shown. *P* values were obtained using the Mann-Whitney U test. (B) Correlation of neutralization score with CD4$^+$ T cell counts in 2009 and 2014. (C) Correlation of neutralization score with patient age in 2009 and 2014. Data for samples collected in 2009 and in 2014 are in blue and in red, respectively. The linear trend is shown with mean and 95% CI bands; Spearman *r* and *P* values are indicated.

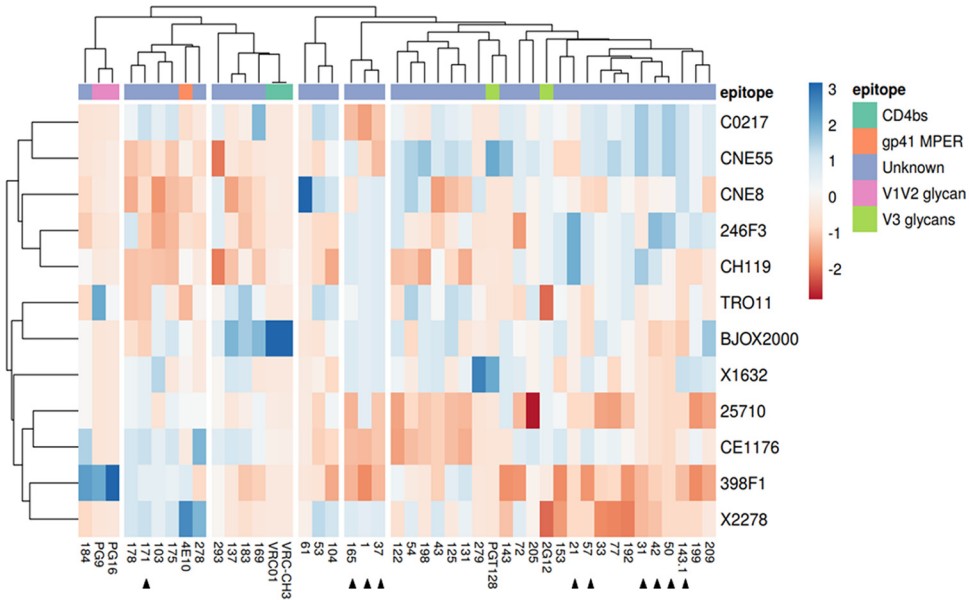

**FIG 6** Cluster analysis and heat map of the predicted epitope specificity in the top neutralizing patients from Angola. At the top of the columns, known bNAb epitopes are colored according to the respective epitope specificities, as shown in the key. The identification of the plasma samples and bNAbs is shown at the bottom. Cluster analysis for both rows and columns was done according to the Pearson correlation (46). Blue colors in the heat map represent lower neutralization activity and red colors higher neutralization activity. Each column represents the neutralization values of a given plasma sample or a bNAb of known specificities against the panel of 12 Env-pseudotyped viruses whose names are on the right. Black triangles indicate samples from 2014.

higher neutralizing-antibody responses (Fig. 7). This was significant for all subtypes of the C2V3C3 recombinant polypeptides tested, confirming this epitope as an important neutralizing domain in these patients independent of subtype.

**Impact of the neutralizing antibodies in the diversity and evolution of C2V3C3.** Neutralizing antibodies targeting the C2, V3 and C3 envelope regions are common in HIV-1-infected individuals (1), and escape from these antibodies leads to higher diversity in these regions as well as to higher positive selection and convergent evolution (50–52). We analyzed amino acid entropy and the sites under selective pressure in the C2, V3, and C3 regions in the different neutralization categories for samples collected in 2009. Considering the three regions together, mean overall entropy was similar in all neutralization categories: weak neutralizers/nonneutralizers, 0.5727 (95% CI, 0.4753 to 0.6241); cross neutralizers, 0.5931 (0.5012 to 0.6849); broad neutralizers, 0.5381 (0.4472 to 0.6291); and elite neutralizers, 0.4106 (0.3313 to 0.4899). Regardless of neutralization category, the region with higher mean entropy was C3 (0.8528 [0.7556 to 0.9500]) followed by V3 (0.4659 [0.3903 to 0.5414]) and C2 (0.3635 [0.3092 to 0.4178]) ($P < 0.0001$). Viruses from broad and cross neutralizers did not vary significantly from viruses from nonneutralizers/weak neutralizers. On the other hand, viruses from elite neutralizers were far less variable than viruses from weak neutralizers/nonneutralizers, as seen by the number of amino acids with positive entropy differences relative to amino acids with negative entropy differences (37 versus 16 sites; $P = 0.0023$) (Fig. 8). The most variable amino acid residues in the broad and elite neutralizers relative to nonneutralizers/weak neutralizers were found in V3 and/or C3 (broad neutralizers, 1 site in C2 versus 7 sites in C3 [$P < 0.0001$]; elite neutralizers, 3 sites in C2 versus 13 sites in V3C3 [$P < 0.0001$]), two regions that contain broadly neutralizing epitopes (Fig. 8).

Diversifying selection in C2V3C3 varied according to the different neutralization categories ($P < 0.001$) (Table S3). Considering only sites that were selected by at least two methods, weak neutralizers/nonneutralizers had a total of 9 positively selected sites, cross neutralizers 6, broad neutralizers 3, and elite neutralizers 1. Regardless of neutralization category most sites under selective pressure were present in the C3 region.

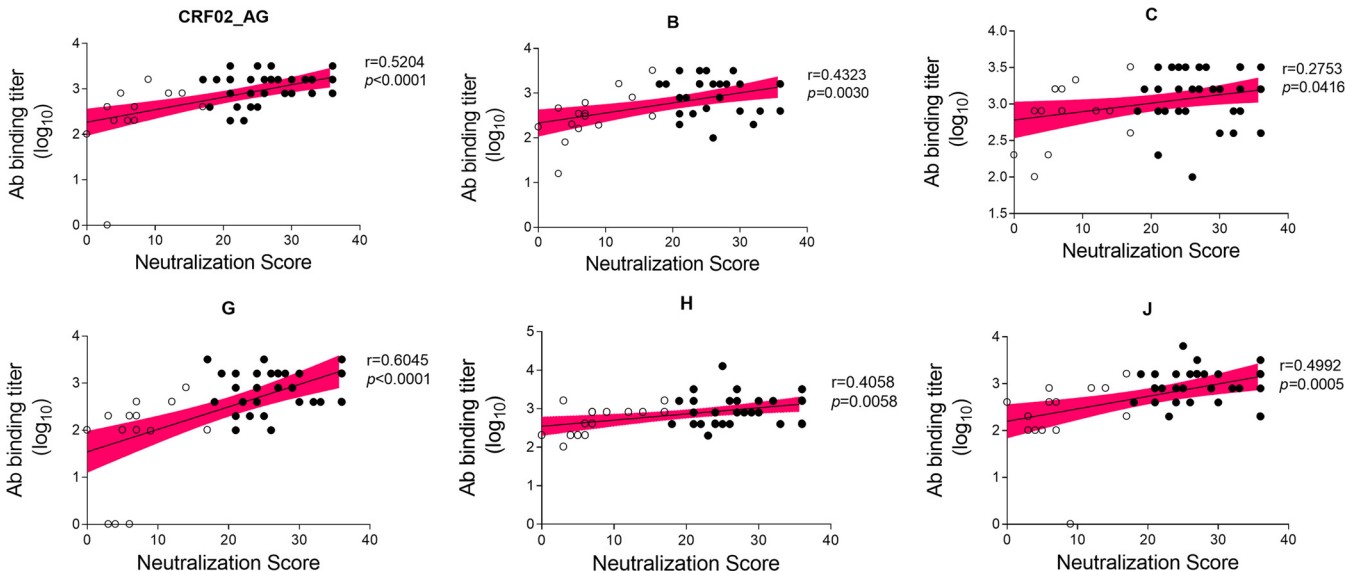

**FIG 7** Association between titer of antibody binding to C2V3C3 recombinant polypeptides of different subtypes and neutralization score (in 2009). Filled symbols represent titers of antibody binding of the broad/elite neutralizers to a given C2V3C3 subtype. Unfilled symbols are the titers of antibody binding of the nonneutralizers/weak neutralizers and cross neutralizers to a given C2V3C3 subtype. Associations were assessed by Spearman analyses. $P$ values and Spearman $r$ values are indicated. Linear trend is shown with mean and 95% CI bands.

The mean number of N-glycosylation sites in C2V3C3 was similar in all neutralization categories (nonneutralizers, 9.2 [range, 8 to 12]; cross neutralizers, 9.2 [range, 7 to 11]; broad neutralizers, 9.4 [range, 7 to 11]; elite neutralizers, 9.8 [range, 8 to 11]) (Table S4). In C3, site 332, which together with site 301 in V3 and other elements in V1, V3, and V4 is part of the V3-glycan supersite (53, 54), was highly conserved (70%) in all neutralization categories.

## DISCUSSION

In this study, we confirmed the extremely high diversity and evolving complexity of HIV-1 strains present in Angola. Subtypes A and C dominated over other subtypes, but all other Env subtypes were present along with untypeable basal strains and recombinant strains that prevailed over pure subtypes. The remarkable diversity and evolution of HIV-1 in Angola is driven by the increasing incidence (55), the limited access to antiretroviral therapy, and high levels of drug resistance (42, 56). The high diversity and rapid evolution of HIV-1 in this country can pose a serious challenge to vaccination and other preventive efforts.

At the individual level, the long-term B cell stimulation by this highly diversified ensemble of viruses may have promoted the development of exceptional neutralization breadth (1, 3, 44, 53, 57–61). We found that the majority (56%) of the patients in our cohort developed cross, broad, or elite neutralizing responses. These results are in line with those of Hraber et al. obtained with samples from chronically infected patients from diverse geographic regions who were infected with diverse HIV-1 subtypes and recombinant forms (7). These authors found that 10% of the samples neutralized >90% of an extensive panel of pseudoviruses (elite neutralizers) and 50% of the samples neutralized 50% of pseudoviruses (broad neutralizers). On the other hand, our results far exceed those from previous cohort studies in sub-Saharan Africa (1, 57, 62–64). For example, Beirnaert et al. found 10.6% broad neutralizers in Cameroon (64), and Landais et al. found about 15% broad neutralizers in a cohort of HIV-1-infected patients from Eastern and South Africa (1). Compared to cohort studies from other countries where subtype B dominates, the frequency of patients with bNAb responses reported in our study was also much higher (3, 4, 65). For example, Rusert et al. (3) in Switzerland found that most patients (79.1%) showed weak or no neutralization breadth, which compares to 44% in our cohort, and that only 1.3% were elite neutralizers, which compares

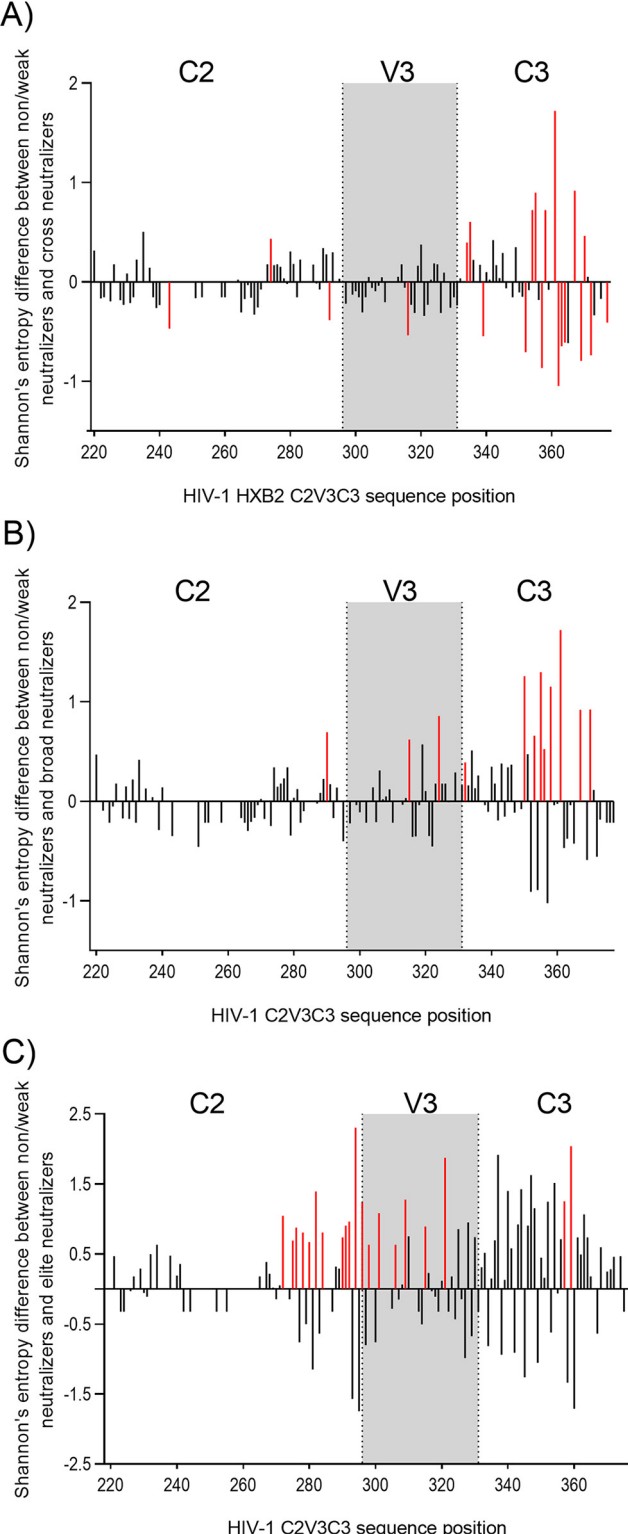

FIG 8 Amino acid entropy differences in the C2V3C3 region between neutralization categories. (A) Shannon's entropy difference between nonneutralizers/weak neutralizers and cross neutralizers. (B) Shannon's entropy difference between nonneutralizers/weak neutralizers and broad neutralizers. (C) Shannon's entropy difference between nonneutralizers/weak neutralizers and elite neutralizers. Sites with significant entropy difference ($P \leq 0.05$) are shown in red. Gray boxes delimit the V3 region. Numbers on the $x$ axes indicate the amino acid position in HIV-1.

to 19% in our cohort. This divergence may be related to many factors besides the diversity of infecting viruses, such as the HLA genotype and ethnicity of the patients, viral load, $CD4^+$ T cell counts, and duration of infection (1, 3, 57, 58, 66). Also, we cannot exclude the role of intrinsic confounders such as assay parameters related to the neutralization assay, e.g., the virus panel and the neutralization categorization methodology.

In agreement with other studies, neutralization score was inversely associated with $CD4^+$ T cell counts in 2009 when the patients were naive to ART and had high viral load (1, 3, 53, 58). This is generally associated with high envelope stimulation of B cells and inevitably leads to B-cell exhaustion in chronic viremic HIV-1 infection (67). On the other hand, the frequency of elite neutralizers and the mean neutralization score increased significantly in 2014, when patients were undergoing ART and had low viral loads. The boost in the quality of the neutralizing response in these patients suggests good restoration of the B cell compartment with ART, which is uncommon in chronic HIV-1 infection (67–69). The moderate level of plasma viremia (median, 11,660 HIV-1 RNA copies/mL; IQR, 380 to 30,060) found in these patients may have provided the low-level antigenic stimulation needed for the full maturation of memory B cells and bNAb production (30, 67, 70). This has a precedent in HIV-2 infection, where most patients are infected for long periods and produce potent and broadly neutralizing responses in a setting of low or undetectable plasma viremia (71–74).

Viral type and subtype as well as the nature of the epitope target on the viral envelope impact the antibody maturation process, as seen by the frequency of elicitation (73, 75) and epitope specificity of bNAbs (3, 43, 54). Differences in envelope structure and epitope exposure, length of variable loops, type of V3 motifs, N-glycosylation patterns, and conservation of key sites have helped to explain why certain HIV-1 subtypes like subtype C are better at promoting the elicitation of neutralizing antibodies (1, 3, 43, 54, 63, 75, 76). In line with these studies, we found that subtype C infection was associated with enhanced neutralization breadth and potency. In general, subtype C infected individuals show a bias toward V2-glycan directed antibody responses, and subtype C envelope from transmitted viruses are less prone to neutralization by V3-directed antibodies due to the absence of the N332 glycan in the C3 region (1, 3, 54, 63, 77–79). This was not the case in our study, as most top neutralizers had V3-directed antibodies and the N301 and N332 glycans defining the V3-glycan supersite were highly conserved in the patients' isolates. Supporting the major role of the V3 and C3 envelope regions in the development of bNAbs in our cohort, we found a strong direct correlation between the titer of antibodies binding to C2V3C3 envelope polypeptides from all subtypes and neutralization score. Nonetheless, antibodies specific for the V2 apex, the CD4 binding site, the gp41 MPER, and/or unknown epitopes were also found in some patients, revealing the complexity of the neutralizing-antibody responses in these patients.

We also looked at the variability of patients' sequences in the envelope C2V3C3 region to assess the impact of escape from neutralizing antibodies on viral evolution and diversity. V3 and C3 were the most variable regions, which is consistent with the dominant role of neutralizing antibodies targeting these regions in these patients. V3 bNAb recognition sites and sites associated with resistance to neutralization, such as N295 (1, 53, 58), were under positive selection in broad and elite neutralizers. However, elite neutralizers exhibited far less variability and lower numbers of sites under selective pressure in V3 and C3 than weak or low neutralizers. The convergence of the viral swarm to a few resistant strains provides convincing evidence for the crucial role of V3- and C3-directed bNAbs in controlling HIV-1 replication and diversification in these patients (63, 75, 80, 81).

In conclusion, an exceptionally high number of Angolan patients infected with HIV-1 produce broad and elite neutralizing antibodies mostly targeting the V3-glycan supersite. A higher prevalence of elite neutralizers was found in patients infected for longer periods and with low viral loads. Our results have direct implications for the development of the next generation of candidate HIV-1 vaccines, as they suggest that

prolonged and low-level V3 and C3 antigenic stimulation from highly diverse isolates favors the elicitation of broadly neutralizing antibodies.

## MATERIALS AND METHODS

**Study population and ethics statement.** This cross-sectional retrospective study included 322 HIV-1-infected adults. Plasma samples were collected in 2001, 2009, and 2014 at the Hospital da Divina Providência (HDP), a referral hospital in Luanda, the capital city of Angola. Eligible participants were ≥19 years of age, were not pregnant, and had a serological diagnosis of HIV-1 (Determine HIV-1/2 [Abbott] and Uni-Gold Recombigen [Trinity Biotech] rapid tests). Plasma viral load and number of CD4$^+$ T cells were determined in a subset of patients using the Abbott real-time HIV-1 assay (Abbott Laboratories) and the ABACUS 5 Junior hematology analyzer, respectively. The study was conducted according to the Declaration of Helsinki and was reviewed and approved by the National Ethics Committee of Angola. The study was verbally explained to all the patients before their written consent was obtained.

**Cell lines.** TZM-bl (RRID, CVCL_B478) and HEK-293T (RRID, CVCL_0063) cells were obtained from the NIH AIDS Reagent Program (https://www.niaid.nih.gov/research/nih-aids-reagent-program). TZM-bl cells were engineered from HeLa cells that constitutively express CXCR4 to express large amounts of CD4, CCR5, and a firefly luciferase reporter gene under the control of the HIV-1 long terminal repeat (LTR) (82). Cells were cultured at 37°C and 5% CO$_2$ using Dulbecco minimal essential medium (DMEM) supplemented with 10% heat-inactivated fetal bovine serum and with 100 U/mL of penicillin and 100 $\mu$g/mL of streptomycin.

**Viral RNA extraction, PCR amplification, sequencing, and phylogenetic analysis.** Viral RNA extraction from plasma was done with a QIAamp viral RNA minikit (Qiagen). Reverse transcription was performed with an NZY first-strand cDNA synthesis kit (NZYtech, Portugal), and a 534-bp fragment comprising the C2V3C3 *env* region was amplified by PCR using an in-house method described elsewhere (39, 83). Sequencing of the C2V3C3 amplicons was performed with a BigDye Terminator cycle sequencing kit (Applied Biosystems). Sequences were aligned with reference strains using Muscle (84). The best-fit model of nucleotide substitution was determined with ModelTest v3.7 (85). Maximum-likelihood trees were inferred with PhyML 3.0 (86). Selective pressure was determined with Datamonkey (https://www.datamonkey.org/) (87). We used four different codon-based maximum-likelihood methods to estimate the $dN/dS$ (also known as $K_a/K_s$ or $\omega$) ratio at every codon in the alignment, including SLAC (single-likelihood ancestor counting), FEL (fixed effects likelihood), IFEL (internal fixed effects likelihood), and REL (random effects likelihood), and sites under selective pressure identified by two or more methods were considered valid.

**Entropy and N-linked-glycosylation analysis.** Potential N-linked glycosylation sites were identified using the N-Glycosite software (88), and the entropy at each amino acid position was measured with Shannon's entropy-one and Shannon's entropy-two online tools, all available at the Los Alamos National Laboratory HIV sequence database (http://www.hiv.lanl.gov/).

**Production of C2V3C3 polypeptides and analysis of antibody reactivity.** Six 178-amino-acid polypeptides comprising part of C2, V3, and part of C3 (positions 212 to 390 in gp120 in HIV-1 HXB2) of HIV-1 isolates circulating in Angola (subtypes C, G, H, J, and CRF02_AG) and Portugal (subtype B) were expressed in *Escherichia coli* and purified, and antibody reactivity against these polypeptides was determined using an enzyme-linked immunosorbent assay (ELISA), as previously described (75).

**Production of Env-pseudotyped viruses.** A reference panel of 12 tier 2 HIV-1 Env-pseudotyped viruses of subtypes C ($n = 3$), A ($n = 1$), CRF07_BC ($n = 2$), CRF01_AE ($n = 2$), B ($n = 2$), G ($n = 1$), and AC recombinant ($n = 1$) were produced using the global panel of HIV-1 Env clones (89), obtained through the NIH AIDS Reagent Program. Env-pseudotyped viruses were produced by transfection of Env-expressing plasmids in 293T cells using pSG3.1Δenv as the backbone in a 1:3 ratio with JetPRIME DNA transfection reagent. Viral stocks were filtered through 0.45-$\mu$m-pore-size filters after 48 h and stored at −80°C until use.

**Plasma neutralization assay.** Neutralization of the Env-pseudotyped viruses was assessed in TZM-bl cells using Tat-induced luciferase (Luc) reporter gene expression to quantify the reduction in virus infection (90). Briefly, TZM-bl cells (10,000 cells/well) were seeded the day before the neutralization assay to allow adherence of the cells to the bottom of the wells. Heat-inactivated plasma samples (56°C for 30 min) were incubated at 1:40 dilution in triplicate with the respective Env-pseudotyped virus (200 TCID$_{50}$/well) for 1 h at 37°C before transfer to TZM-bl cells. After 48 h, percent neutralization was determined by calculating the difference in average relative light units (RLU) between test wells containing plasma samples and the wells containing the Env-pseudotyped virus from the indicator panel after the normalization of the results using the average RLU of cell control wells. Results were considered valid if the average RLU of virus wells was >10 times the average RLU of cell control wells. A virus pseudotyped with the envelope glycoprotein of vesicular stomatitis virus (VSV-G) was used as a neutralization specificity control. Plasma samples with the capacity to inhibit VSV-G were excluded.

Neutralizing-antibody titers were determined for a subset of plasma samples with known antibody profiles ($n = 64$). In this case, 100 $\mu$L of 2-fold serial dilutions beginning at 1:40 were mixed with 100 $\mu$L of each virus (200 50% tissue culture infective doses [TCID$_{50}$]/well) and incubated for 1 h before being added to the cells. After 48 h, culture medium was removed from each well, and plates were analyzed for luciferase activity as described above. Wells with medium were used as background controls, and virus-cell wells were included as infection controls. Neutralizing titer (ID$_{50}$) was defined as the highest dilution for which 50% neutralization was achieved.

**Neutralization score and plasma categorization.** The percent neutralization for each plasma-virus combination was recorded as a breadth-potency matrix: ≥80% neutralization received a score of 3, 50% to <80% a score of 2, 20% to <50% a score of 1, and <20% a score of 0. Plasma samples were then ranked by the sum of scores in order to reflect their potency and breadth (3, 62). As a validated and worldwide accepted classification system to define neutralizing activity is lacking, for the purpose of the present study we classified plasma samples with scores 25 to 36 as elite neutralizers,18 to 24 as broad neutralizers, 6 to 17 as cross neutralizers, and <6 as weak neutralizers or nonneutralizers. According to this classification, a plasma sample from an elite neutralizer must neutralize ≥9 viruses of the panel with a neutralization potency of ≥80%.

**Prediction of bNAb epitope specificities by clustering analysis.** The neutralizing antibody specificities were determined for a subset of patients exhibiting broad and elite neutralization capacity using clustering analysis with human bNAbs targeting the main neutralizing epitopes on the viral envelope and able to neutralize at least half of the 12 Env-pseudotyped viruses of the panel, as described previously (89). Neutralization heat maps and clusters were computed via the online tool ClustVis using a predefined correlation clustering distance method (Pearson correlation subtracted from 1) based on the average distance of all possible pairs. ClustVis is a web tool for visualizing clustering of multivariate data (available at https://biit.cs.ut.ee/clustvis/) (46).

**Statistical analysis.** The Mann-Whitney, Kruskal-Wallis, and Fisher's exact tests were used to compare differences between groups. The Spearman rank test was used to quantify the magnitude and direction of the correlation between antibody neutralization activity and plasma binding titers against C2V3C3 polypeptides, $CD4^+$ T cell counts, viral subtype, and age of patients. Hypothesis tests were two-tailed, and $P$ values of <0.05 were considered significant. To test the potential correlation between neutralization score and genetic distance of the clinical samples to the neutralization panel viruses, we used amino acid sequences and Hamming distances that included gaps as characters, because (i) neutralization occurs on the amino acid level, (ii) neutralization does not depend on the evolutionary path to a state combination, and (iii) indels may have significant effects on antibody binding. Genetic distances were calculated using DECIPHER (91), regression analysis was performed using R version R-4.0.3 (92), and visualization was carried out using ggplot2 (93).

**Data availability.** Sequences produced in this work were given GenBank accession numbers OM960847 to OM960948, OP485752 to OP485758, OM960847 to OM960847, OP485752 to OP485758, AY456278 to AY456308, AY676573, AY676574, AY676576 to AY676586, AY684933, EU031840 to EU031884, and EU031886 to EU031891. Deidentified participant data collected for this study will be shared upon request.

## SUPPLEMENTAL MATERIAL

Supplemental material is available online only.

**SUPPLEMENTAL FILE 1**, PDF file, 1.4 MB.

## ACKNOWLEDGMENTS

We gratefully acknowledge the contribution and efforts of the staff and patients from the Hospital da Divina Providência in Luanda for this study. TZM-bl cells were obtained through the NIH AIDS Reagent Program, Division of AIDS, NIAID, NIH.

This work was supported by ADEIM-FFUL (Associação para o Ensino e a Investigação em Microbiologia) and by Fundação para a Ciência e a Tecnologia (FCT), Portugal, under project grants UIDB/04138/2020 and UIDP/04138/2020. This study was in part supported by the NIH/NIAID under grant R01AI087520. F.M. was supported by FCT under PhD grant number SFRH/BD/87488/2012. I.B. and C.P. are funded by FCT under a contract program as defined by DL no. 57/2016 and law no. 57/2017. The funding source was not involved in study design, in the collection, analysis, and interpretation of data, in the writing of the report, or in the decision to submit the paper for publication.

N.T. and S.C. conceived and designed the study. S.C. provided the patients' data and specimens. F.M., J.M.M., I.B., S.T., I.M., P.G., R.M., R.C., P.B., and T.L. performed the experiments. F.M., J.M.M., C.P., T.L., and N.T. analyzed the data. F.M., T.L., P.B., and N.T. performed the statistical analysis. F.M., T.L., P.B., and N.T. drafted the manuscript and discussed the final version. All authors read and approved the final manuscript.

We report no conflicts of interests.

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
