## [Reviewer comments · Microbiology Spectrum]

Microbiology Spectrum

Long-term and low-level envelope C2V3 stimulation from highly diverse virus isolates leads to frequent development of broad and elite antibody neutralization in HIV-1 infected individuals

Francisco Martin, José Marcelino, Claudia Palladino, Inês Bártolo, Susana Tracana, Inês Moranguinho, Paloma Gonçalves, Rita Mateus, Rita Calado, Pedro Borrego, Thomas Leitner, Sofia Clemente, and Nuno Taveira

Corresponding Author(s): Nuno Taveira, Faculdade de Farmácia, Universidade de Lisboa

Review Timeline:

Submission Date:	May 9, 2022
Editorial Decision:	June 29, 2022
Revision Received:	October 16, 2022
Accepted:	October 29, 2022

Editor: Yongjun Sui

Reviewer(s): The reviewers have opted to remain anonymous.

Transaction Report:

DOI: <https://doi.org/10.1128/spectrum.01634-22>

June 29, 2022

Prof. Nuno Taveira
Faculdade de Farmácia, Universidade de Lisboa
iMed.Ulisboa
Avenida Prof Gama Pinto
Lisboa 1649-003
Portugal

Re: Spectrum01634-22 (Long-term and low-level envelope C2V3 stimulation from highly diverse virus isolates leads to frequent development of broad and elite antibody neutralization in HIV-1 infected individuals)

Dear Prof. Nuno Taveira:

Link Not Available

Sincerely,

Yongjun Sui

Journals Department
Reviewer comments:

Reviewer #1 (Public repository details (Required)):

Some HIV sequences were obtained and it seems authors provided accession numbers although not sure if all were included as seen in my comments to the authors.

Reviewer #1 (Comments for the Author):

In this manuscript Martin et al. describe an in-depth analysis of the neutralization profiles of plasma samples obtained from a

cohort of individuals living with HIV in Angola. The main findings of the study are that more than half of the individuals presented with broadly neutralizing responses, the relevance of the cohort and the reasonable correlations found: longer infection time correlated with better neutralization, more binding antibodies in plasma meant better neutralization, clade C infection correlated with better plasma activity, etc. I found the manuscript well written and the conclusions clear. The strengths of the manuscript are the exhaustive analysis of the data; that the findings are logical and the number and relevance of the samples tested. Weaknesses of the manuscript are that some of the analysis rely on software predictions and that it was difficult to follow at times due to wrong order of figures.

I have the following comments for the authors:

- One of the main findings of the study is that the majority of patients (56%) made broadly neutralizing responses. Authors compare their findings with other studies that found 10.6% or 15%, but how does their criteria for 'broadly neutralizing responses' compare to the criteria used in those other studies? In line with this, did the other studies use potency as % of neutralization at a lower or higher plasma dilution instead the 1:40 used here? or did they define breadth as number of viruses neutralized more than 20% like in this study, or was the cutoff higher or lower? Did those studies use similar viruses or other panel?
- Can the authors rule out that the increase in neutralization in samples from 2014 is not due to interference from ART that may be present in the plasma? I wonder whether the higher neutralization activity found in 2014 vs 2019 depends on whether the individuals started taking ART.
- In many instances the manuscript calls out the wrong figure, which made it confusing:
 - *Line 347: I believe Fig. S5 should be called out instead of S3.
 - *Lines 349: idem as above.
 - *Line 245: Fig. S2 seems incorrectly called here. Should that be Fig. 1?
 - *Figure S6 does not seem to be called out at all.
 - *Page 37-38:
 - Line 797 legend reads Fig. S2 but I believe this corresponds to Fig. S3.
 - Line 804 legend reads Fig. S3 but I believe this corresponds to Fig. S5.
 - Line 809 legend reads Fig. S4 but I believe this corresponds to Fig. S6.
 - *Figure S4:
 - Description in legend for panel A seems to actually belong to panel C.
 - Description in legend for panel B seems to actually belong to panel A.
 - Description in legend for panel C seems to actually belong to panel B.
- How does exactly ClustVis predict neutralizing epitope specificities? Please explain for clarity.
- Line 73: I believe more than five vulnerability sites have now been reported, please update.
- Line 137: please check numbers, it seems only 101 sequences were uploaded
- Line 251: I think it would be useful to add some explanation here about the NS or at least refer to the corresponding methods section.
- Fig. 2B: brown and red color where a bit difficult to differentiate for me.
- Some figures could benefit from a legend (example Fig. 5B & C and S3) and increased font size in axis (example: Fig S5).
- Figure 3A: I found confusing that there are two columns with 'clade' here. I would change the first 'clade' to 'individual's clade' (or the like) and the second 'HIV clade' to 'HIV-1 clade and env-pseudotyped virus tested' or the like. In this same panel (3A) authors should consider adding a legend for the colors used or at least indicate in the figure's legend what's the threshold for each color code. Ex: orange is 80 to 160, etc. In this figure 'NA' is not defined in the legend. Please add.
- Figure 4: could the light gray squares in two of the p-values be removed?
- Figure S5A: Please indicate in legend type of analysis/graph this is and software used to generate it.
- Line 378. How does this prediction exactly work and how robust is the epitope specificity prediction of the plasma vs bNAbs? Please comment on it and add references.
- Line 397: I believe authors meant 'to the right' and not 'to the left'.

- Line 405: It seems Figure S5 was cited before Figure S4 (provided S3 was incorrect and should have been S5).
- Figure 8: I recommend adding C2 and C3 as well to the graphs (in the likes of V3).
- Table S3: I found it quite confusing. Many abbreviations were not defined, such as REL, PP, FEL, etc. Some explanation in the text of the manuscript or legend/notes in the table is highly advisable.
- Figure S2: For enhanced clarity, authors should explain a bit more about the tools used to generate the calculations shown in this figure (like what's the geno2pheno and the Shannon's entropy). There is a typo: 'determine' should read 'determined'.

Reviewer #2 (Public repository details (Required)):

HIV-1 Env partial sequences

Reviewer #2 (Comments for the Author):

In this manuscript entitled, "Long-term and low-level envelope C2V3 stimulation from highly diverse virus isolates leads to frequent development of broad and elite antibody neutralization in HIV-1 infected individuals", submitted to Microbiology Spectrum by Nuno Taveira and colleagues, the authors assessed the neutralizing activity of 322 HIV-infected individuals in Angola.

In summary, the work is methodologically correct and important for the HIV field, since there are no many reports of this kind with an important number of tested subjects. Moreover, the results are in line with previous report from Hraber and colleagues, adding more information about this particular subject. However, the authors should adjust their main conclusions according to the data presented here and discuss it in the context of previous reports, properly.

MAJOR COMMENTS related to the authors' main conclusions are:

1. 52% of individuals developed cross neutralizing antibodies (NAbs). They used TZM-bl assay and a panel of 12 tier 2 pVs to make this particular analysis. The authors suggest that this proportion is higher than that reported in the literature. I agree with that partially. Although most estimations of breadth and neut potency in the population are generally lower it is important to note that most studies of this type are biased due to limitations related to the estimation approach (i.e., NEUT ASSAY, panel of viruses used, etc.). To my knowledge one of the most controlled study in this regard is the one by Hraber et al. AIDS. 2014 January 14; 28(2): 163-169. doi:10.1097/QAD.000000000000106. In that study the authors tested more than 200 samples from diverse geographical regions vs a virus panel composed by >200 tier2 pV. In that study, the authors observed that 10% of samples neutralized >90% pVs (elite) and 50% samples neutralized 50% of pVs (broad). The authors should adjust their conclusions based on this previous report in all sections of the present manuscript (abstract, intro, results, discussion and conclusion)

2. Broad NAb response was associated with longer infection time/ duration, HIV-1 Subtype C infection, lower CD4 Tcell counts, higher age, level of C2V3C3-sp Abs.

Regarding LONGER INFECTION TIME: although there is evidence in the literature that supports the fact that Ab affinity maturation that occurs during the course of infection correlates with breadth and neut potency, the authors do not account for a substantial number of sequential paired samples to support this observation. They have a limited number of samples (FIG2.C) that they used to sequentially measure NAbs in 2009 and 2014. The authors should make emphasis on their own results and explain what are the limitations of their particular study regarding this point, then they can discuss this finding in the context of supportive literature. For example in Line 296-298: "Overall, these results suggest that duration of infection is an important correlate of the potency and breadth of the neutralizing antibody response in Angolan patients infected with HIV-1". Unmatched sequential samples do not allow to make this observation. Again, most results pointed out in lines 268-295, correspond to unmatched samples. Sequential samples from same individuals along the course of infection are necessary to support these statements.

Regarding LOWER CD4 CELL COUNT: this result is somewhat contradictory analysing the data presented here.

Line 359-366: "Drug naïve patients (year 2009) with {less than or equal to}200 CD4+ T cells/µl at study entry had significantly higher NS values than patients with >200 CD4+ T cells/µl [median NS in patients {less than or equal to}200 CD4+ T cell counts was 7.00 (IQR, 3.50, 21.00) vs 4.00 (1.00, 12.00) in patients with > 200 CD4+ T cell counts, p = 0.0193] (Figure 5A). Moreover, NS values were inversely associated with CD4+ T cell counts (Spearman r=-0.3043, p=0.0005) (Figure 5B) and directly associated with age (Spearman r=0.1644, p =0.0302) in these patients (Figure 5C). These results suggest that elicitation of high levels of broadly neutralizing antibodies in these patients is directly correlated with prolonged antigenic stimulation"

Line 225-227: "The median number of CD4+ T cells in 2014 was 1.8-fold higher when compared to 2009 (p=0.0015). The lower VL and higher CD4+ T cell number in 2014 is consistent with most patients being on cART which was not the case in 2001 and 2009" and Line 280: "In line with the previous results, neutralizing score increased in 2014 relative to 2009 in 31 out of the 38

matched plasma pairs analysed (81.6%)"

The authors discuss these contradictory results in the discussion; Line 484-494: "In agreement with other studies, neutralization score was inversely correlated with CD4+ T cell counts in 2009 when the patients were naïve to ART and had high viral load. This is generally associated with high envelope stimulation of B cells and inevitably leads to B-cell exhaustion in chronic viraemic HIV-1 infection. Remarkably, however, the frequency of elite neutralizers and the mean neutralization score increased significantly in 2014, when patients were already undergoing ART, relative to 2009. The boost in the quality of the neutralizing response in these patients suggest good restoration of the B cell compartment with ART which is uncommon in chronic HIV-1 infection. The moderate level of plasma viremia (median 11,660 HIV-1 RNA copies /ml, IQR, 380-30,060) found in these patients may have provided the low-level antigenic stimulation needed for the full maturation of memory B cells and bNAbs production" Again I feel that the main problem here is making conclusions from different datasets that are not linked (i.e., samples from 2009 and 2014). The negative correlation (NeutScore vs CD4) observed in 2009 is weak (0.3) and then they compared what happened in 2009 vs 2014 (CD4 counts, Viremia, Neutralization Score, cART, etc). The absence of sequential paired samples assessing PVL, CD4, NEUT and cART status (coverage and adherence) does not allow making strong conclusion at this point. In consequence, the statements regarding CD4/PVL/cART/NS are blurry.

3. Main target of NABs was V3-gly supersite. In addition, C2-V3-C3 region was less variable in elite vs weak neutralizers, suggesting a key role of these particular NABs in HIV control. This observation was based on an in silico approach comparing plasma neut profiles with those observed for prototypic mAbs (i.e., 2G12, PGT128, etc), an ELISA assay with particular polypeptides and an aminoacid entropy analysis. Although the authors observed a positive correlation of binding to C2-V3-C3 peptides from different clades and neut score, the binding of polyclonal Abs present in these samples to other Env-peptides was not tested and can't be excluded. This limitation must be stated in the discussion.

Regarding variability, the authors acknowledge previous studies showing that breadth might be influenced by higher number of quasispecies circulating within infected individuals (Line 104-106: "The genetic complexity of the HIV-1 quasispecies present in infected individuals is directly related to the development of neutralization breadth regardless of infection duration (refs40,41)"). How can the authors explain lower variability of Env V3-C3 in the best controllers? In which way the Env sequencing approach might be biasing this observation; for example, batch sequence vs single genome (SGA) vs NGS and the fact of not accounting for full Env sequences.

Regarding the Shannon entropy plots: Could you please explain what negative values mean (L441 "amino acids with negative entropy"? Since I am used to see only positive values on them. Could you please explain how do you determine lower variability in the ELITE vs WEAK (L440-442)? Since it looks more variable to me.

4. Long-term low-level V3-C3 stimulation in these individuals promotes elicitation of bNABs. Again regarding the long-term is an assumption not fully supported by current data. Moreover, in Line 78-80 the authors mention: "An exception is the V3-glycan supersite bNAb lineage that does not require extensive antibody-affinity maturation allowing their development in early stages of infection, and explaining their high prevalence in recently infected individuals", which is somewhat contradictory to previous statement.

In second place I am not convinced of the term "low-level" used by the authors since they did not performed a particular analysis aimed to linking viral loads and neut score in these population. Although mean viremia was lower in 2014 compared to previous time-points the lack of a comprehensive sequential analysis invalidates this assumption.

Minor comments:

Line 74: should be "V1V2 apex".

Line 165: "...at 1:40 dilution in triplicate with the respective Env-pseudotyped virus..." indicate TCID50 as you did below (titration assay).

Line 221-222. "The median age of the patients was 34 years and most (n=242, 64.5%) were women. The main route of transmission was heterosexual contact (n=304, 81.1%)." Proportions should be calculated using 322 as denominator since you are talking about patients characteristics (not samples).

Line 218-219: "Overall, 375 plasma samples from 322 adult HIV-1 infected patients from three sampling years, 2001 (n=106), 2009 (n=210) and 2014 (n=59)" vs Line 243-245: "A total of 236 plasma samples, 178 from 2009 and 58 from 2014, were screened for neutralization breadth and potency against the 12 Env-pseudotyped indicator panel, amounting to 2832 plasma/virus combinations (Figure S2)." Explain the reasons why not all available samples were tested for NEUT. Avoid "amounting to 2832 plasma/virus combinations"

Line 247-250: "Likewise, the mean percent neutralization..." should be "the overall neutralization potency calculated as the % neut at fix dilution 1:40 vs all tested variants", for example. Fix (27.43%, 95%CI: 25.94, 28.92 in 2009 vs 60.52%, 95%CI: 57.37, 63.66 in 2014, p<0.0001) and (4.39, 95%CI: 3.83, 4.95 in 2009 vs 8.40, 95%CI: 7.32, 9.48 in 2014, p<0.0001) to: (27.43%; CI95: 25.94-28.92 in 2009 vs 60.52%, CI95: 57.37-63.66 in 2014; p<0.0001).

Line 252-253: Remarkably, approximately 30% (n= 68/236) of the patients developed antibody responses with the capacity to potentially neutralize at least half the viruses from the panel. This statement is not clearly sustained by that figure. 236 are samples, not patients.

Line 319-320: "These results indicate that virus subtype is a major determinant of the neutralizing antibody response in our patients." ...For this particular population at this particular time-point.

Line 481: "...divergence may be related to..." Also assay parameters i.e., NEUT assay, pVs panel, etc

Line 528-529: "These results have direct implications for the development of the next-generation of HIV-1 vaccines" For example?

Staff Comments:

Preparing Revision Guidelines

Please return the manuscript within 60 days; if you cannot complete the modification within this time period, please contact me. If you do not wish to modify the manuscript and prefer to submit it to another journal, please notify me of your decision immediately so that the manuscript may be formally withdrawn from consideration by Microbiology Spectrum.

Responses to Reviewers

Re: Spectrum01634-22 (Long-term and low-level envelope C2V3 stimulation from highly diverse virus isolates leads to frequent development of broad and elite antibody neutralization in HIV-1 infected individuals)

Reviewer #1 (Public repository details (Required)):

Some HIV sequences were obtained and it seems authors provided accession numbers although not sure if all were included as seen in my comments to the authors.

The GenBank accession numbers of all sequences produced in this work are now provided in the paper.

Reviewer #1 (Comments for the Author):

In this manuscript Martin et al. describe an in-depth analysis of the neutralization profiles of plasma samples obtained from a cohort of individuals living with HIV in Angola. The main findings of the study are that more than half of the individuals presented with broadly neutralizing responses, the relevance of the cohort and the reasonable correlations found: longer infection time correlated with better neutralization, more binding antibodies in plasma meant better neutralization, clade C infection correlated with better plasma activity, etc. I found the manuscript well written and the conclusions clear.

The strengths of the manuscript are the exhaustive analysis of the data; that the findings are logical and the number and relevance of the samples tested. Weaknesses of the manuscript are that some of the analysis rely on software predictions and that it was difficult to follow at times due to wrong order of figures.

We thank the reviewer for the positive appreciation of our work. Regarding the number of figures, they are mentioned correctly in the paper. However, six supplementary figures were originally uploaded incorrectly as the submitted paper only cites four supplementary figures. In addition, the number indicated in the inset legend of some of the supplementary figures was incorrect. We have now corrected both these mistakes.

I have the following comments for the authors:

- One of the main findings of the study is that the majority of patients (56%) made broadly neutralizing responses. Authors compare their findings with other studies that found 10.6% or 15%, but how does their criteria for 'broadly neutralizing responses' compare to the criteria used in those other studies? In line with this, did the other studies use potency as % of neutralization at a lower or higher plasma dilution instead the 1:40 used here? or did they define breadth as number of viruses neutralized more than 20% like in this study, or was the cutoff higher or lower? Did those studies use similar viruses or other panel?

We are glad that you alluded to this fact. Indeed, a standardized and well-established way to categorize the neutralization profile of HIV infected patients is not yet available

which complicates study comparison. We used a standardized neutralization assay in TZM-bl cells and a reference and global panel of 12 HIV-1 pseudoviruses (<https://www.hiv.lanl.gov/content/nab-reference-strains/html/home.htm>). To categorize the neutralization profile, we used the neutralization score described by Rusert et al (2016) which considers both the neutralization potency (% neutralization) and breadth (number of Env-pseudotyped viruses neutralized). There is no formal recommendation to use a determined plasma dilution for the screening of neutralizing activity in human plasma samples. Three reference studies performed with many samples from different geographical regions, including Africa, used different experimental conditions. Rusert et al. (2016) used a 1/150 plasma dilution and a 8-virus panel for the initial screening of patients with neutralizing response. Hraber et al., AIDS (2014), used three-fold plasma dilutions starting at 1/20 and a 219-virus panel. Landais et al (2016) used three-fold plasma dilutions starting at 1/100 and a 6-virus panel. All studies, including ours, determined the neutralization titers (ID50) of the plasma samples showing broad and elite neutralization capacity in the initial screening which confirmed the neutralizing activity at higher plasma dilution in most patients. We therefore believe that it is reasonable to compare our results with these other studies. However, we recognize that an additional effort is required to standardize neutralization studies performed all over the world so that results from these studies can be directly compared. In this sense, we think that the neutralization score, and neutralization assay and reference isolates that we have used should be considered the gold-standard.

- Can the authors rule out that the increase in neutralization in samples from 2014 is not due to interference from ART that may be present in the plasma? I wonder whether the higher neutralization activity found in 2014 vs 2009 depends on whether the individuals started taking ART.

We thank the reviewer for this question. After analyzing the results of the neutralization activity in 2014 and comparing them with the results obtained in 2009, the significant differences observed between years drew our attention to this issue. However, the impact of ART in neutralization was excluded due to the negative neutralization results obtained with the neutralization specificity control which is an HIV-1 strain pseudotyped with the envelope glycoprotein of vesicular stomatitis virus (VSV-G). This control lacks the HIV env gene but has a fully functional and wild-type (i.e., without drug resistant mutations) pol gene meaning that it is fully susceptible to the antiretroviral drugs used in Angola. If ART was active, it would inhibit the replication of the VSV-G control pseudovirus which was not the case in almost all our plasma samples. In fact, plasma samples with the capacity to inhibit the VSV-G pseudovirus were excluded from further analysis. This information was added to the revised manuscript in lines 193-194.

- In many instances the manuscript calls out the wrong figure, which made it confusing:

The reviewer is right, and we apologize for the mistake in uploading more than the cited supplementary figures and in the numeration of some supplementary figures (see above). This has been corrected.

*Line 347: I believe Fig. S5 should be called out instead of S3.

The reviewer is right, and we apologize for the mistake in uploading more than the cited supplementary figures and in the numeration of some supplementary figures (see above). This has been corrected.

*Lines 349: idem as above.

The reviewer is right, and we apologize for the mistake in uploading more than the cited supplementary figures and in the numeration of some supplementary figures (see above). This has been corrected.

*Line 245: Fig. S2 seems incorrectly called here. Should that be Fig. 1?

Both Figure 1 and Figure S2 should be mentioned. This was indicated in line 269 of the revised manuscript.

*Figure S6 does not seem to be called out at all.

The reviewer is right, and we apologize for the mistake in uploading more than the cited supplementary figures and in the numeration of some supplementary figures (see above). This has been corrected.

*Page 37-38:

Line 797 legend reads Fig. S2 but I believe this corresponds to Fig. S3.

The legend is correct and now corresponds to the right supplementary figure.

Line 804 legend reads Fig. S3 but I believe this corresponds to Fig. S5.

The legend is correct and now corresponds to the right supplementary figure.

Line 809 legend reads Fig. S4 but I believe this corresponds to Fig. S6.

The legend is correct and now corresponds to the right supplementary figure.

*Figure S4:

Description in legend for panel A seems to actually belong to panel C.

Description in legend for panel B seems to actually belong to panel A.

Description in legend for panel C seems to actually belong to panel B.

The legend is correct and now corresponds to the right supplementary figure.

- How does exactly ClustVis predict neutralizing epitope specificities? Please explain for clarity.

For delineating antibody specificities of the HIV-1 neutralizing polyclonal Angolan plasma samples we used a clustering method based on the neutralizing ability (as determined by the IC50s) of reference broadly neutralizing monoclonal antibodies (bNAbs) that target the main neutralizing epitopes of the HIV-1 envelope and that neutralize at least half of the ENV-pseudotyped virus of our panel. In this method clustering of the plasma samples with the monoclonal antibodies implies that the sample has antibodies recognizing the bNAb epitope. Our input data for this analysis consisted of an Excel table with the IC50 values of our subset of 38 plasma samples from broad and elite neutralizers together with IC50 values of 7 bNabs of known epitope specificities, against the 12 ENV-pseudotyped viruses of the indicator panel. ClustVis, a web tool for visualizing clustering of multivariate data (available at <https://biit.cs.ut.ee/clustvis/>), was used for this analysis. Cluster analysis for both rows and columns were computed according to the Pearson correlation. Clusters and neutralization heatmaps are formed based on the average distance of all possible pairs (virus and polyclonal or monoclonal antibodies). The method is described in Metsalu et al. 2015 (our ref 53). Doria-Rose et al used a similar clustering method to assess breadth of human immunodeficiency virus-specific neutralizing activity in sera (Doria-Rose et al., JVI 2010; PMID: 19923174).

- Line 73: I believe more than five vulnerability sites have now been reported, please update.

We have made minor modifications in this paragraph because the major sites of bNAbs vulnerability in the envelope glycoproteins of HIV-1 are still only five (Lines 73-76). On this issue please see the recent review of Haynes et al. in Nature Review Immunology (2022).

- Line 137: please check numbers, it seems only 101 sequences were uploaded

GenBank accession numbers of all sequences have now been mentioned.

- Line 251: I think it would be useful to add some explanation here about the NS or at least refer to the corresponding methods section.

For clarity we added a note referring to the methods section in line 277.

-Fig. 2B: brown and red color where a bit difficult to differentiate for me.

Increased contrast was added to facilitate differentiation.

- Some figures could benefit from a legend (example Fig. 5B & C and S3) and increased font size in axis (example: Fig S5).

Font size of Figure S3 (originally Fig. S5) was increased. Legends of Figure S3 is given at the end of the paper and in the figure itself. Legend to Figure 5B and C is given in the text.

- Figure 3A: I found confusing that there are two columns with 'clade' here. I would change the first 'clade' to 'individual's clade' (or the like) and the second 'HIV clade' to 'HIV-1 clade and env-pseudotyped virus tested' or the like. In this same panel (3A) authors should consider adding a legend for the colors used or at least indicate in the figure's legend what's the threshold for each color code. Ex: orange is 80 to 160, etc.

In this figure 'NA' is not defined in the legend. Please add.

Thank you for your suggestions. We have changed figure 3A accordingly. The heatmap color code and the meaning of NA were added to the legend.

- Figure 4: could the light gray squares in two of the p-values be removed?

Sorry we cannot see any light gray squares in the p-values.

- Figure S5A: Please indicate in legend type of analysis/graph this is and software used to generate it.

This is now Figure S3 and the required information was added to the legend.

- Line 378. How does this prediction exactly work and how robust is the epitope specificity prediction of the plasma vs bNAbs? Please comment on it and add references.

There is no ideal method to determine the neutralizing epitopes in plasma from HIV infected patients. The clustering method we have used should be considered an approximation to the identification of the neutralizing epitopes. Antigen adsorption methods and use of Env-pseudotyped viruses with mutations disrupting the main neutralizing epitopes are experimental strategies used in other studies. All these methods have limitations given the complexity of some of the neutralizing epitopes in HIV. As mentioned above for delineating antibody specificities of the HIV-1 neutralizing polyclonal Angolan plasma samples we used a clustering method based on the neutralizing ability (as determined by the IC50s) of reference broadly neutralizing monoclonal antibodies (bNAb) that target the main neutralizing epitopes of the HIV-1 envelope and that neutralize at least half of the ENV-pseudotyped virus of our panel. In this method clustering of the plasma samples with the monoclonal antibodies implies that the sample has antibodies recognizing the bNAb epitope. Our input data for this analysis consisted of an Excel table with the IC50 values of our subset of 38 plasma samples from broad and elite neutralizers together with IC50 values of 7 bNabs of known epitope specificities, against the indicator panel of 12 ENV-pseudotyped virus. ClustVis, a web tool for visualizing clustering of multivariate data (available at <https://biit.cs.ut.ee/clustvis/>), was used for this analysis. Cluster analysis for both rows and columns were computed according to the Pearson correlation. Clusters and neutralization heatmaps are formed based on the average distance of all possible pairs (virus and polyclonal or monoclonal antibodies). The method is described in Metsalu et al. 2015 (our ref 53). Doria-Rose et al used a similar clustering method to assess breadth of human immunodeficiency virus-specific neutralizing activity in sera (Doria-Rose et al., JVI 2010; PMID: 19923174).

- Line 397: I believe authors meant 'to the right' and not 'to the left'.

Yes, thank you for spotting that out. We have changed the legend of the figure accordingly (line 433)

- Line 405: It seems Figure S5 was cited before Figure S4 (provided S3 was incorrect and should have been S5).

Sorry but figure S5 e Fig. S3 as mentioned above.

- Figure 8: I recommend adding C2 and C3 as well to the graphs (in the likes of V3).

Done as suggested.

- Table S3: I found it quite confusing. Many abbreviations were not defined, such as REL, PP, FEL, etc. Some explanation in the text of the manuscript or legend/notes in the table is highly advisable.

This issue was clarified in the Methods section, lines 157 to 161.

- Figure S2: For enhanced clarity, authors should explain a bit more about the tools used to generate the calculations shown in this figure (like what's the geno2pheno and the Shannon's entropy). There is a typo: 'determine' should read 'determined'.

There is a misunderstanding here related with a mistake in the inset legend of the original Figure S2. As indicated in the legend to the correct figure S2 in lines 975-981, this figure is a heatmap showing the neutralizing activity of plasma samples from 2009 and 2014.

Reviewer #2 (Public repository details (Required)):

HIV-1 Env partial sequences

Reviewer #2 (Comments for the Author):

In this manuscript entitled, "Long-term and low-level envelope C2V3 stimulation from highly diverse virus isolates leads to frequent development of broad and elite antibody neutralization in HIV-1 infected individuals", submitted to Microbiology Spectrum by Nuno Taveira and colleagues, the authors assessed the neutralizing activity of 322 HIV-infected individuals in Angola.

In summary, the work is methodologically correct and important for the HIV field, since there are no many reports of this kind with an important number of tested subjects. Moreover, the results are in line with previous report from Hraber and colleagues, adding more information about this particular subject. However, the authors should adjust their main conclusions according to the data presented here and discuss it in the context of previous reports, properly.

MAJOR COMMENTS related to the authors' main conclusions are:

1. 52% of individuals developed cross neutralizing antibodies (NAbs). They used TZM-bl assay and a panel of 12 tier 2 pVs to make this particular analysis. The authors suggest that this proportion is higher than that reported in the literature. I agree with that partially. Although most estimations of breadth and neut potency in the population are generally lower it is important to note that most studies of this type are biased due to limitations related to the estimation approach (i.e., NEUT ASSAY, panel of viruses used, etc.). To my knowledge one of the most controlled study in this regard is the one by Hraber et al. AIDS. 2014 January 14; 28(2): 163-169. doi:10.1097/QAD.000000000000106. In that study the authors tested more than 200 samples from diverse geographical regions vs a virus panel composed by >200 tier2 pV. In that study, the authors observed that 10% of samples neutralized >90% pVs (elite) and 50% samples neutralized 50% of pVs (broad). The authors should adjust their conclusions based on this previous report in all sections of the present manuscript (abstract, intro, results, discussion and conclusion)

We thank the reviewer for pointing out this important publication that, sadly, we have missed. The paper was adjusted as indicated.

2. Broad NAb response was associated with longer infection time/ duration, HIV-1 Subtype C infection, lower CD4 Tcell counts, higher age, level of C2V3C3-sp Abs.

Regarding LONGER INFECTION TIME: although there is evidence in the literature that supports the fact that Ab affinity maturation that occurs during the course of infection correlates with breadth and neut potency, the authors do not account for a substantial number of sequential paired samples to support this observation. They have a limited number of samples (FIG2.C) that they used to sequentially measure NAbs in 2009 and 2014. The authors should make emphasis on their own results and explain what are the limitations of their particular study regarding this point, then they can discuss this finding in the context of supportive literature. For example in Line 296-298: "Overall, these results suggest that duration of infection is an important correlate of the potency and breadth of the neutralizing antibody response in Angolan patients infected with HIV-1". Unmatched sequential samples do not allow to make this observation. Again, most results pointed out in lines 268-295, correspond to unmatched samples. Sequential samples from same individuals along the course of infection are necessary to support these statements.

It is true that we have a limited number of matched samples and that we must have this in mind when taking any conclusions. However, neutralizing score increased in 2014

relative to 2009 in almost all matched plasma pairs analysed (31/38, 81.6%) and this was highly significant ($P < 0.0001$). We therefore think that it is reasonable to conclude that duration of infection has contributed to increase potency and breadth of neutralizing response in our cohort. Nonetheless, to be less assertive about this and eliminate the word correlate which is formally incorrect, we changed the position and words of the final sentence to:

"The significant increase in neutralizing score in 2014 relative to 2009 in matched samples suggest that higher duration of infection contributes to increase the potency and breadth of the neutralizing antibody response in Angolan patients infected with HIV-1."

This sentence was placed after the description of results presented in Figure 2.

Regarding LOWER CD4 CELL COUNT: this result is somewhat contradictory analysing the data presented here.

Line 359-366: "Drug naïve patients (year 2009) with {less than or equal to}200 CD4+ T cells/ μ l at study entry had significantly higher NS values than patients with >200 CD4+ T cells/ μ l [median NS in patients {less than or equal to}200 CD4+ T cell counts was 7.00 (IQR, 3.50, 21.00) vs 4.00 (1.00, 12.00) in patients with > 200 CD4+ T cell counts, $p = 0.0193$] (Figure 5A). Moreover, NS values were inversely associated with CD4+ T cell counts (Spearman $r = -0.3043$, $p = 0.0005$) (Figure 5B) and directly associated with age (Spearman $r = 0.1644$, $p = 0.0302$) in these patients (Figure 5C). These results suggest that elicitation of high levels of broadly neutralizing antibodies in these patients is directly correlated with prolonged antigenic stimulation"

Line 225-227: "The median number of CD4+ T cells in 2014 was 1.8-fold higher when compared to 2009 ($p = 0.0015$). The lower VL and higher CD4+ T cell number in 2014 is consistent with most patients being on cART which was not the case in 2001 and 2009" and Line 280: "In line with the previous results, neutralizing score increased in 2014 relative to 2009 in 31 out of the 38 matched plasma pairs analysed (81.6%)"

The authors discuss these contradictory results in the discussion; Line 484-494: "In agreement with other studies, neutralization score was inversely correlated with CD4+ T cell counts in 2009 when the patients were naïve to ART and had high viral load. This is generally associated with high envelope stimulation of B cells and inevitably leads to B-cell exhaustion in chronic viraemic HIV-1 infection. Remarkably, however, the frequency of elite neutralizers and the mean neutralization score increased significantly in 2014, when patients were already undergoing ART, relative to 2009. The boost in the quality of the neutralizing response in these patients suggest good restoration of the B cell compartment with ART which is uncommon in chronic HIV-1 infection. The moderate level of plasma viremia (median 11,660 HIV-1 RNA copies /ml, IQR, 380-30,060) found in these patients may have provided the low-level antigenic stimulation needed for the full maturation of memory B cells and bNAbs production"

Again I feel that the main problem here is making conclusions from different datasets that are not linked (i.e., samples from 2009 and 2014). The negative correlation (NeutScore vs CD4) observed in 2009 is weak (0.3) and then they compared what happened in 2009 vs 2014 (CD4 counts, Viremia, Neutralization Score, cART, etc). The absence of sequential paired samples assessing PVL, CD4, NEUT and cART status (coverage and adherence) does not allow making strong conclusion at this point.

In consequence, the statements regarding CD4/PVL/cART/NS are blurry.

Data from our patients from 2009 shows a direct association of neutralization score with age and an inverse association (not correlation) with the number of CD4+T cells. Association and correlation are very different things and so we have revised the discussion replacing “correlated” by “associated”. The remainder of the discussion in this part was clarified so as not to link the datasets and prevent unsupported conclusions. We agree with the reviewer that a better discussion would have been possible if the samples were paired but unfortunately that was not the case for most of them. In addition, information on CD4 numbers and viral load was incomplete in most samples from 2014 preventing meaningful comparisons of these parameters and NS in the matched samples from 2009. This data discontinuity is one of the problems when working with patients/samples from this part of Africa. Be as it may, it is certain that ART increases the number of CD4+T cells which may help B cells to produce high levels of neutralizing antibodies as observed in these patients. We think that the changes made in the text clarify this point and prevent unsupported and abusive conclusions.

3. Main target of NAb was V3-gly supersite. In addition, C2-V3-C3 region was less variable in elite vs weak neutralizers, suggesting a key role of these particular NAb in HIV control. This observation was based on an *in silico* approach comparing plasma neut profiles with those observed for prototypic mAbs (i.e., 2G12, PGT128, etc), an ELISA assay with particular polypeptides and an aminoacid entropy analysis. Although the authors observed a positive correlation of binding to C2-V3-C3 peptides from different clades and neut score, the binding of polyclonal Abs present in these samples to other Env-peptides was not tested and can't be excluded. This limitation must be stated in the discussion.

We agree with this comment, and this was made very clear in the original discussion as follows: “Supporting the major role of the V3 and C3 envelope regions in the development of bNAb in our cohort, we found a strong direct correlation between the titer of antibodies binding to C2V3C3 envelope polypeptides from all subtypes and neutralization score. Nonetheless, antibodies specific for the V2 apex, the CD4 binding site, the gp41 MPER and/or unknown epitopes were also found in some patients revealing the complexity of the neutralizing antibody responses in these patients.”

Regarding variability, the authors acknowledge previous studies showing that breadth might be influenced by higher number of quasispecies circulating within infected individuals (Line 104-106: "The genetic complexity of the HIV-1 quasispecies present in infected individuals is directly related to the development of neutralization breadth regardless of infection duration (refs40,41)") . How can the authors explain lower variability of Env V3-C3 in the best controllers? In which way the Env sequencing approach might be biasing this observation; for example, batch sequence vs single genome (SGA) vs NGS and the fact of not accounting for full Env sequences.

In the best controllers neutralizing antibodies will contribute to control virus replication. With limited virus replication there is lower amino acid diversity in the C2-V3-C3 region and lower ability to escape neutralization. The prevailing viruses in these patients should be the few that are able to replicate in the presence of the neutralizing antibodies. So, in the few HIV patients that develop potent and broadly neutralizing antibody responses there is an initial period where the higher number and complexity

of quasispecies promotes the development of neutralization breadth which is followed by a period of inhibition of virus replication by these bNAbs and by the antiretroviral drugs they might be taking which limits virus evolution and diversity in the plasma.

We agree that our sequencing approach gives a limited view of the envelope variability and that sequencing the full Env genes and using deeper sequencing approaches such as the ones mentioned by the reviewer could have provided additional layers of complexity to our data.

Regarding the Shannon entropy plots: Could you please explain what negative values mean (L441 "amino acids with negative entropy"? Since I am used to see only positive values on them. Could you please explain how do you determine lower variability in the ELITE vs WEAK (L440-442)? Since it looks more variable to me.

We calculated the entropy difference between Weak/no neutralizers and the other neutralization categories for each amino acid. Each time the entropy, i.e., the variability of the amino acid sequence, is higher in the Weak/no neutralizers the result is positive and when the difference is significant the bars are shown in red. When the entropy value is higher for the elite category the difference gives a negative value and when this is significant the bars are shown in red. Hence, red positive bars signal amino acids where the entropy is higher in the weak neutralizers and red negative bars signals amino acids with higher entropy in the other categories. It is immediately apparent that the number of red positive bars is much higher than the number of red negative bars when comparing weak with elite neutralizers meaning that the overall entropy is much higher in the weak than in the elite neutralizers.

4. Long-term low-level V3-C3 stimulation in these individuals promotes elicitation of bNAbs. Again regarding the long-term is an assumption not fully supported by current data. Moreover, in Line 78-80 the authors mention: "An exception is the V3-glycan supersite bNAb lineage that does not require extensive antibody-affinity maturation allowing their development in early stages of infection, and explaining their high prevalence in recently infected individuals", which is somewhat contradictory to previous statement.

This conclusion is mostly based on the neutralization results obtained in matched patients. As mentioned in the paper, neutralizing score (NS) increased in 2014 relative to 2009 in 31 out of the 38 matched plasma pairs analysed (81.6%). Hence, as mentioned above, we think that it is reasonable to conclude that duration of infection and lower viral load contributed to the increase in NS in these patients. Regarding the sentence in line 78-80 there is no contradiction in the results because the analysis of epitope specificity was mostly done in samples from 2009 (28 samples out of 38 analysed) and of the 22 samples that clustered with PGT128 and 2G12 most (16, 72.7%) were from 2009. In addition, bNAbs targeting the V3-mannose patch belong to several lineages and while some do not require extensive affinity maturation (as better explained in the revised introduction) others, like PGT128, do require extensive affinity maturation and do not appear in the early stages of infection.

In second place I am not convinced of the term "low-level" used by the authors since they did not performed a particular analysis aimed to linking viral loads and neut score in these population. Although mean viremia was lower in 2014 compared to previous

time-points the lack of a comprehensive sequential analysis invalidates this assumption.

The neutralizing score increased significantly from 2009 to 2014 in matched patients and the prevalence of elite neutralizers was much higher in 2014. This occurred in a setting of low viral load due to antiretroviral therapy and implies low level antigenic stimulation in these patients. We therefore think that it is reasonable to associate low level antigenic stimulation for prolonged periods of time with increased ability to elicit broadly neutralizing antibodies. The conclusion was modified to convey this message better.

Minor comments:

Line 74: should be "V1V2 apex".

Corrected.

Line 165: "...at 1:40 dilution in triplicate with the respective Env-pseudotyped virus..." indicate TCID50 as you did below (titration assay).

Done

Line 221-222. "The median age of the patients was 34 years and most (n=242, 64.5%) were women. The main route of transmission was heterosexual contact (n=304, 81.1%)." Proportions should be calculated using 322 as denominator since you are talking about patients characteristics (not samples).

Thank you for pointing it out. We changed the text accordingly (Lines 244-245): "The median age of the patients was 34 years and most (n=242, 75.2%) were women. The main route of transmission was heterosexual contact (n=304, 94.4%)."

Line 218-219: "Overall, 375 plasma samples from 322 adult HIV-1 infected patients from three sampling years, 2001 (n=106), 2009 (n=210) and 2014 (n=59)" vs Line 243-245: "A total of 236 plasma samples, 178 from 2009 and 58 from 2014, were screened for neutralization breadth and potency against the 12 Env-pseudotyped indicator panel, amounting to 2832 plasma/virus combinations (Figure S2)." Explain the reasons why not all available samples were tested for NEUT. Avoid "amounting to 2832 plasma/virus combinations"

Thank you for the opportunity to clarify. The amount of plasma that we had for certain individuals was very limited preventing their use in the neutralization studies. That is the reason behind the differences between Lines 218-219 and Lines 243-245. We have decided to remove the sentence "amounting to 2832 plasma/virus combinations" as per your recommendation.

Line 247-250: "Likewise, the mean percent neutralization..." should be "the overall neutralization potency calculated as the % neut at fix dilution 1:40 vs all tested variants", for example. Fix (27.43%, 95%CI: 25.94, 28.92 in 2009 vs 60.52%, 95%CI: 57.37, 63.66 in 2014, p<0.0001) and (4.39, 95%CI: 3.83, 4.95 in 2009 vs 8.40, 95%CI: 7.32, 9.48 in 2014, p<0.0001) to: (27.43%; CI95: 25.94-28.92 in 2009 vs 60.52%, CI95: 57.37-63.66 in 2014; p<0.0001).

The sentence was changed as suggested.

Line 252-253: Remarkably, approximately 30% (n= 68/236) of the patients developed antibody responses with the capacity to potently neutralize at least half the viruses from the panel. This statement is not clearly sustained by that figure. 236 are samples, not patients.

We rephrase the sentence according to your comment: "Remarkably, approximately 30% (n= 68/236) of the plasma samples tested from the Angolan patients had antibody responses with the capacity to potently neutralize at least half the viruses from the panel (Figure 2A and 2B)."

Line 319-320: "These results indicate that virus subtype is a major determinant of the neutralizing antibody response in our patients." ...For this particular population at this particular time-point.

We rephrase the sentence according to your comment: "These results indicate that in 2009 virus subtype was a major determinant of the neutralizing antibody response in our HIV-1 infected Angolan patients"

Line 481: "...divergence may be related to..." Also assay parameters i.e., NEUT assay, pVs panel, etc

According to your suggestion we added: Also, differences between studies such as assay parameters related to the neutralization assay e.g., the virus panel, the neutralization categorization methodology, cannot be exclude.

Line 528-529: "These results have direct implications for the development of the next-generation of HIV-1 vaccines" For example?

According to our results it is important to develop vaccines that enable a durable low-level immune stimulation with Envelope based antigens of increasing diversity to promote the development of broadly neutralizing antibodies.

October 29, 2022

Prof. Nuno Taveira
Faculdade de Farmácia, Universidade de Lisboa
iMed.Ulisboa
Avenida Prof Gama Pinto
Lisboa 1649-003
Portugal

Re: Spectrum01634-22R1 (Long-term and low-level envelope C2V3 stimulation from highly diverse virus isolates leads to frequent development of broad and elite antibody neutralization in HIV-1 infected individuals)

Dear Prof. Nuno Taveira:

Your manuscript has been accepted, and I am forwarding it to the ASM Journals Department for publication. You will be notified when your proofs are ready to be viewed.

Sincerely,

Yongjun Sui
Editor, Microbiology Spectrum
